# Uni-Synergy: Bridging Understanding and Generation for Personalized Reasoning via Co-operative Reinforcement Learning

## Abstract

Unified Multimodal Models (UMMs) excel in general tasks but struggle to bridge the gap between personalized understanding and generation. Prior works largely rely on implicit token-level alignment via supervised fine-tuning, which fails to fully capture the potential synergy between comprehension and creation. In this work, we propose Sync-R1, an end-to-end reinforcement learning framework that jointly optimizes personalized understanding and generation within a single, explicit reasoning loop. Through this unified feedback process, Sync-R1 enables personalized comprehension to guide content creation, while the resulting generation quality reciprocally refines understanding within an integrated reward landscape. To efficiently orchestrate this dual-task synergy, we introduce Sync-GRPO, a reinforcement learning method utilizing an ensemble reward system. Furthermore, we propose Dynamic Group Scaling (DGS), which adaptively filters low-potential trajectories to reduce gradient variance and accelerate convergence. To better reflect real-world complexity, we introduce UnifyBench++, featuring denser textual descriptions and richer user contexts. Experimental results demonstrate that Sync-R1 achieves state-of-the-art performance, showcasing superior cross-task reasoning and robust personalization without requiring complex cold-start procedures.

## 1. Introduction

The rapid evolution of Unified Vision-Language Models (ULM) (Wu et al., 2024b; Jiang et al., 2025; Team, 2025; Xie et al., 2025a; Li et al., 2025; Xie et al., 2025b; Deng et al., 2025) has established a powerful paradigm for general-purpose multimodal AI, yet the potential of unified models to execute complex, personalized reasoning remains largely under-explored. Real-world tasks typically demand a complex interplay between understanding user-specific contexts and generating personalized content. Existing approaches often rely on implicit token-level sharing or simplistic multi-task supervised fine-tuning (SFT) (Nguyen et al., 2025; An et al., 2025b). These approaches either treat understanding and generation as disjoint objectives (Tian et al., 2025) or demand high-cost training pipelines that fail to capture the deep synergistic relationship between tasks. Consequently, two critical challenges remain: (1) How to establish an explicit mechanism where personalized understanding and generation mutually enhance one another, and (2) How to achieve such coordination efficiently within a unified framework.

To address these challenges, we propose **Sync-R1**, an end-to-end reinforcement learning framework that achieves explicit and synergistic co-optimization of personalized understanding and generation. Unlike previous methods that rely on latent embedding alignment (An et al., 2025b), we bridge the task gap by orchestrating a unified reasoning trajectory (Penha et al., 2025): the model first extracts fine-grained conceptual information from user-provided context (*understanding*) to guide the subsequent creation of personalized content (*generation*). Crucially, this trajectory is optimized holistically under a composite reward function that evaluates both the fidelity of the understanding output and the quality of the final generation. This ensures that the feedback from generation substantially informs and improves the understanding process. Thus, we establish a reciprocal feedback loop where comprehension provides the semantic foundation for generation, and generation provides the objective signal to calibrate comprehension.

While the synergistic potential between understanding and generation is well-recognized, effectively coordinating these two distinct reasoning objectives necessitates a sophisticated optimization strategy. To this end, we propose **Sync-GRPO**, a reinforcement learning method designed to jointly optimize both tasks within a unified loop. We advocate for RL over supervised fine-tuning (SFT) based on two principal considerations: **(1) Eliciting Latent Synergy:** Instead of re-

[1]Anonymous Institution, Anonymous City, Anonymous Region, Anonymous Country. Correspondence to: Anonymous Author <anon.email@domain.com>.

Preliminary work. Under review by the International Conference on Machine Learning (ICML). Do not distribute.

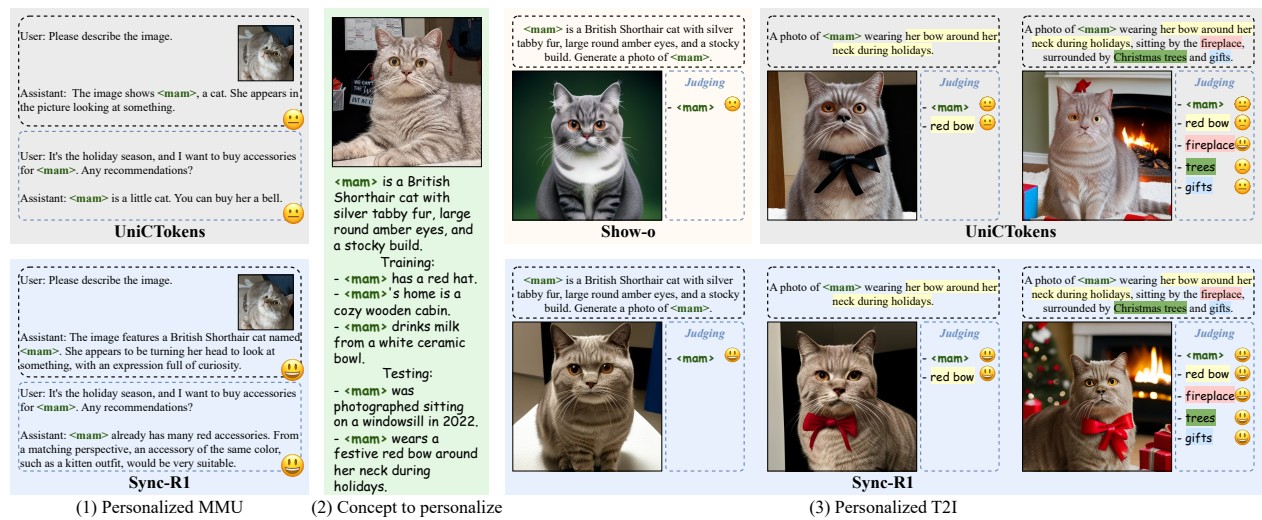

*Figure 1.* **Capability Overview of Sync-R1.** Our framework jointly optimizes personalized understanding and generation within a unified reasoning trajectory via Sync-GRPO. By establishing an explicit synergistic loop, Sync-R1 leverages the reciprocal enhancement between comprehension and creation to achieve precise integration of personalized concepts. Beyond standard personalization, Sync-R1 excels in challenging scenarios, including dense-text and reasoning-based generation tasks with missing concept information.

dundantly re-teaching skills via SFT, RL is uniquely suited to elicit and coordinate the robust foundational capabilities already inherent in UMMs (Chen et al., 2025; Liang et al., 2025). This approach enables the discovery of optimal integration strategies through adaptive self-improvement. **(2) Enhancing Coherent Reasoning:** Leveraging RL's proven efficacy in complex reasoning (Bai et al., 2022; Hu et al., 2025), we design rewards to explicitly incentivize logical alignment between understanding and generation. This steers the model toward coherent, contextually grounded personalization.

To address the prohibitive computational cost of multimodal RL, we further introduce **Dynamic Group Scaling (DGS)**. This adaptive sampling mechanism dynamically prunes low-potential trajectories in the initial stages of generation, substantially reducing gradient variance and accelerating convergence while maintaining high sample quality for reward estimation. Moreover, to establish a more rigorous evaluation benchmark for personalized reasoning, we extend UnifyBench (An et al., 2025b) to **UnifyBench++**, enriching it with denser textual descriptions and more diverse user contexts. Extensive experiments demonstrate that Sync-R1 achieves state-of-the-art performance in all the tasks, exhibiting robust cross-task synergy without requiring costly cold-start fine-tuning.

Our primary contributions are summarized as follows:

- We propose Sync-R1, a unified RL framework establishing an explicit reasoning loop to synergistically co-optimize personalized understanding and generation.
- We develop Sync-GRPO and DGS to effectively and effi-

ciently coordinate the joint optimization of personalized understanding and generation tasks.
- We introduce UnifyBench++, an enhanced benchmark for robust evaluation. Extensive experiments show that Sync-R1 sets a new state of the art, demonstrating superior cross-task synergy without costly cold-start procedures.

## 2. Related Work

**Reinforcement Learning for Multimodal Tasks.** Reinforcement Learning (RL) has established itself as a pivotal paradigm for enhancing model adaptability and complex task-specific reasoning (Yuan et al., 2025; Yue et al., 2025). In visual generation, RL is increasingly explored to transcend the limitations of supervised fine-tuning. Early efforts integrated Chain-of-Thought (CoT) reasoning into the generation process (Chern et al., 2025; Pan et al., 2025b), while the advent of Group Relative Policy Optimization (GRPO) (DeepSeek-AI et al., 2025) has catalyzed efficient RL-driven visual synthesis (Tong et al., 2025; Han et al., 2025; Xue et al., 2025; Xiao et al., 2025). However, applying RL to unified multimodal models (Mao et al., 2025; Zhang et al., 2025b)—specifically to foster explicit synergy between comprehension and creation—remains largely unexplored. Existing personalization methods, in contrast, often rely on isolated optimization or implicit cross-task transfer. Our work represents a pioneering effort to bring this synergistic RL paradigm into the personalization domain. We propose **Sync-R1**, which orchestrates personalized understanding and generation within an explicit, closed-loop reasoning trajectory.

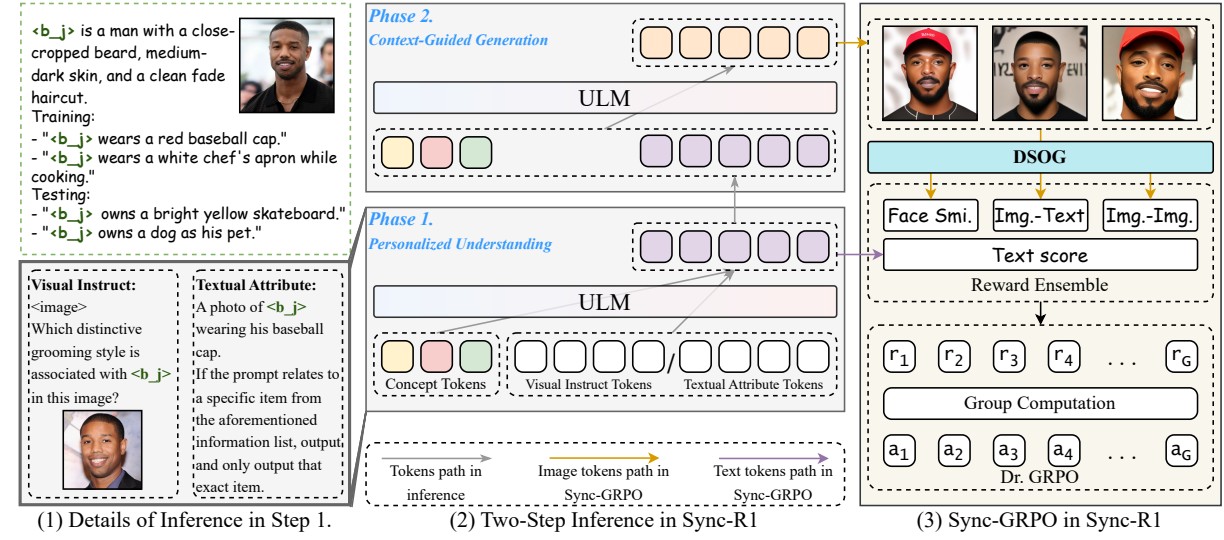

(1) Details of Inference in Step 1.     (2) Two-Step Inference in Sync-R1     (3) Sync-GRPO in Sync-R1

*Figure 2.* **Overview of the Sync-R1 Framework.** We introduce a novel integration of personalized understanding and generation reasoning. Through our proposed **Sync-GRPO**, these two processes are coupled to mutually reinforce each other, jointly contributing to parameter updates. This mechanism significantly enhances the model's reasoning capabilities regarding personalized concepts and achieves high-fidelity information injection.

## 3. Method

To facilitate the explicit synergy between personalized understanding and generation, we propose Sync-R1 (as shown in Figure 2), an end-to-end GRPO reinforcement learning framework, which establishes an explicit reasoning loop where comprehension and creation are jointly optimized within a unified trajectory. In this section, we first define the dual-task personalized reasoning process, then introduce the Sync-GRPO optimization objective, and finally present the Dynamic Group Scaling (DGS) strategy designed to accelerate convergence through gradient variance reduction.

### 3.1. Explicit Synergistic Reasoning

Existing works (Nguyen et al., 2025; An et al., 2025b) typically achieve cross-task information sharing at the token level via multi-task supervised fine-tuning. However, this implicit alignment often creates semantic bottlenecks when processing complex, personalized contexts. To overcome this, we decompose the personalized task into a two-phase explicit reasoning trajectory as detailed below:

**Phase I: Personalized Understanding.** We treat personalized understanding not as a simple classification or captioning task, but as a reasoning process that internalizes personalized concept knowledge from heterogeneous inputs. We define two distinct reasoning pathways as depicted in Figure 2(1): **(1) Visual Instruct Reasoning:** Given a reference image $I$ and a descriptive query $Q$, the model distills a reasoning result $IR$ that captures the fine-grained

attributes of the concept: $I + Q \xrightarrow{\text{UMM}} IR_{vis}$. **(2) Textual Attribute Reasoning:** The model receives all extra attribute information and a reasoning prompt $IP$ from UnifyBench++ (see Appendix B for details). The model must then output the attribute that best conforms to the prompt: $Extra\_info + IP \xrightarrow{\text{UMM}} IR_{txt}$.

**Phase II: Context-Guided Generation.** Based on the reasoning results $IR$ from Phase I, we construct a Compound Prompt $CP$ for each pathway to supervise the image generation process, ensuring a direct semantic flow from comprehension to creation. **(1) For Visual Instruct Reasoning:** We combine a basic prompt $BP$ (e.g., "a photo of $\langle sks \rangle$", where $\langle sks \rangle$ a learnable unique identifier initialized via supervised fine-tuning.) with the visual reasoning result $IR_{vis}$ to form the compound prompt: $CP_{vis} = BP \oplus IR_{vis}$. Here, $IR_{vis}$ enriches the generation with fine-grained visual attributes (e.g., "$\langle sks \rangle$ has a clean fade haircut"), explicitly guiding the generation. **(2) For Textual Attribute Reasoning:** We concatenate the original inference prompt $IP$ (e.g., "a photo of $\langle sks \rangle$ wearing his baseball cap") with the textual reasoning result $IR_{txt}$ (e.g., "$\langle sks \rangle$ wears a red baseball hat") to form the compound prompt: $CP_{txt} = IP \oplus IR_{txt}$. This allows the model to resolve ambiguities or missing details in $IP$ by incorporating the explicitly reasoned attribute $IR_{txt}$. This two-phase decomposition forms an explicit reasoning loop, directly optimizable via reinforcement learning. By designing a reward function that jointly assesses $IR$ and the final generation (Section 3.2), we enable the synergistic co-optimization of comprehension and creation.

## 3.2. Sync-GRPO

To optimize the explicit reasoning trajectory defined in Section 3.1, we introduce **Sync-GRPO**, an objective designed to close the feedback loop between understanding and generation. Our approach builds upon the Group Relative Policy Optimization (GRPO) framework (DeepSeek-AI et al., 2025), eliminating the value function dependency to enable joint optimization across discrete modalities. However, unlike pure text models, our multi-modal backbone (Show-o (Xie et al., 2025a)) employs a discrete diffusion mechanism based on MaskGIT (Chang et al., 2022). In this paradigm, image generation is formulated as a non-autoregressive mask-and-predict process. Starting from a fully masked state $I_T$, the model iteratively predicts and refines the entire set of image tokens over $T$ denoising steps to reach the final image $I_0$:

$$P_\theta(I_0 \mid I_T, CP) \propto \prod_{t=1}^{T} \prod_{k=1}^{N} \pi_\theta(I_{t-1,k} \mid I_t, CP) \quad (1)$$

where $N$ is the total number of image tokens, and $\pi_\theta(I_{t-1,k} \mid I_t, CP)$ represents the probability distribution over all tokens at step $t$. While only masked tokens are updated in each step during inference, the reinforcement learning objective considers the predictions across all positions as meaningful policy decisions that contribute to the final image quality.

We formally model the complete synergistic trajectory as $o_i = (IR_i, \{I_{i,t}\}_{t=0}^{T})$. To ensure stable optimization across varying task difficulties and prevent length-bias from degrading generative quality, we adopt the *Dr.GRPO* formulation (Liu et al., 2025b). The probability ratio $D_{i,j}(\theta)$, which serves as the fundamental learning signal, is defined to span the entire loop:

$$D_{i,j}(\theta) = \begin{cases} \dfrac{\pi_\theta(IR_{i,j} \mid q, IR_{i,<j})}{\pi_{\theta_{\text{old}}}(IR_{i,j} \mid q, IR_{i,<j})}, j \in \text{Tokens in } IR_i \\ \dfrac{\pi_\theta(I_{i,t-1,k} \mid I_{i,t}, CP_i)}{\pi_{\theta_{\text{old}}}(I_{i,t-1,k} \mid I_{i,t}, CP_i)}, j \in \text{Image Tokens} \\ \text{at step } t, \text{ index } k \end{cases} \quad (2)$$

In this formulation, $q$ is determined by the randomly selected pathway (Visual Instruct or Textual Attribute), and $k$ denotes the spatial index within the image token grid at denoising step $t$. To balance the contribution of reasoning versus generation, we assign separate weights $\alpha_{\text{text}}$ and $\alpha_{\text{image}}$ to the two components. The final objective (detailed in Appendix C) maximizes the clipped advantage estimated from this joint ratio while penalizing the KL-divergence from a reference policy for stability. This synergistic loop effectively aligns both modalities, ensuring that the reasoning outputs $IR$ are functionally optimal for guidance, while the generative feedback reciprocally grounds and refines the model's understanding capabilities.

## 3.3. Dynamic Group Scaling

**Motivation.** The performance of Sync-GRPO is intrinsically tied to the group size $G$, but a larger $G$ demands prohibitively long training times. Inspired by selective sampling paradigms (Shrivastava et al., 2025), we propose Dynamic Group Scaling (DGS), which evaluates multiple trajectory candidates and retains only the most promising ones through early-stage screening. This strategy allows the model to concentrate its gradient computation on the most informative regions of the trajectory space, effectively achieving the variance-reduction benefits of an enlarged candidate pool at a fraction of the full-generation cost.

**Method.** As illustrated in Figure 3, we exploit the observation that semantic structures often emerge in early denoising stages (An et al., 2025a). Specifically, at $t \approx 0.2T_{total}$, we compute a surrogate reward $\tilde{R}$ (e.g., the BLIP Evaluation Reward shown at Section 4.2). DGS iteratively samples trajectories until it obtains $G$ successful candidates that satisfy the threshold criterion $\tilde{R}(o_i) > T$. By only completing the generation process for these $G$ candidates, DGS ensures that the resulting gradients are derived from high-potential, semantically grounded samples.

**Theoretical Analysis.** The efficacy of DGS stems from its systematic reduction of gradient estimator variance. Let the terminal reward $R$ and surrogate $\tilde{R}$ for each trajectory be i.i.d. and follow a bivariate normal distribution with correlation $\rho = \text{Corr}(R, \tilde{R}) > 0$.

**Theorem 3.1** (Variance Reduction). *Under a linearized policy gradient regime (omitting clipping and KL terms for analysis), we have:*

$$\text{Tr}(\text{Var}[\nabla_\theta \mathcal{J}]) \propto \text{Var}[\hat{A}] \quad (3)$$

*where $\nabla_\theta \mathcal{J}$ is the standard Sync-GRPO gradient using $G$ random samples and:*

$$\hat{A} = R - \text{E}[R] \quad (4)$$

*If we denote $N$ as the total number of the evaluated trajectories, the trace of the gradient covariance for DGS-enhanced Sync-GRPO satisfies:*

$$\frac{\text{Tr}(\text{Var}[\nabla_\theta \mathcal{J}^{(DGS)}])}{\text{Tr}(\text{Var}[\nabla_\theta \mathcal{J}])} \leq 1 - \rho^2(1 - \frac{G}{N}) \quad (5)$$

Theorem 3.1 implies that DGS suppresses gradient noise by a factor proportional to the filtration intensity $1 - \frac{G}{N}$ and the surrogate fidelity $\rho$. This leads to an amplification of the Stochastic Gradient Signal-to-Noise Ratio (SNR), given by $\text{SNR}(\theta) = \|\mathbb{E}[\mathbf{g}]\|^2 / \text{Tr}(\text{Var}[\mathbf{g}])$ for a gradient estimator $\mathbf{g}$.

**Corollary 3.2** (SNR Amplification). *Based on Theorem 3.1, the SNR of the DGS update relates to the baseline as:*

$$\frac{\text{SNR}^{(DGS)}(\theta)}{\text{SNR}(\theta)} > \frac{1}{1 - \rho^2(1 - \frac{G}{N})}. \quad (6)$$

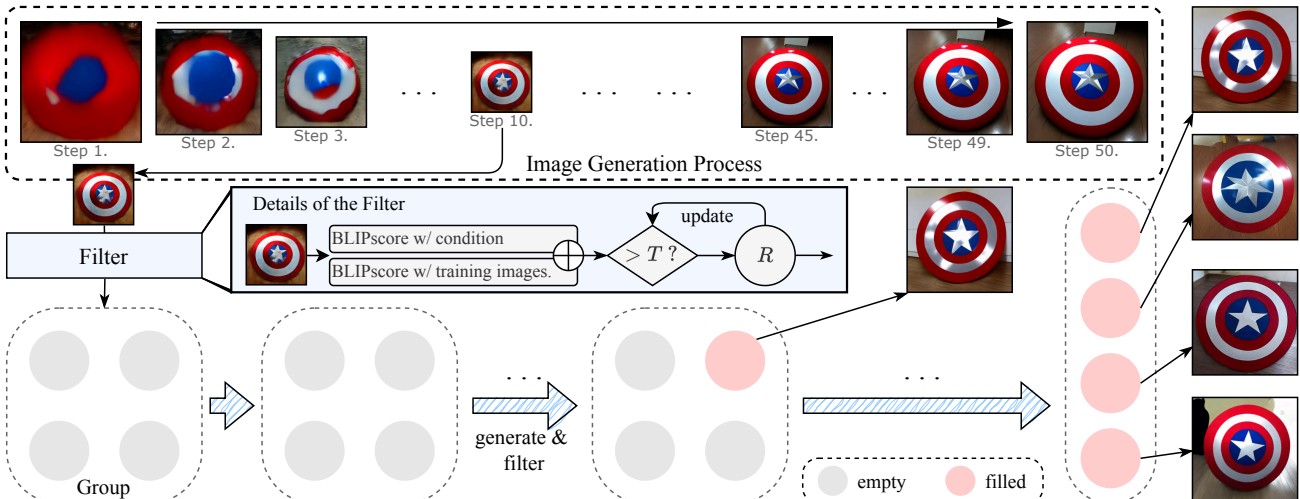

*Figure 3.* **Illustration of Dynamic Group Scaling (DGS).** To improve efficiency, we employ a preliminary assessment at the 10th step of the Show-o generation process. The generation continues only if the intermediate score exceeds a dynamically updated threshold. This threshold is adjusted adaptively to maintain the probability of high-quality generation selection at a target level, effectively filtering out suboptimal trajectories early.

Higher SNR translates to more reliable update directions, accelerating traversal across the optimization manifold. Relative proofs are provided in Appendix D.

**Adaptive Threshold Control.** To maintain a stable selection ratio near the Target Pass Rate (TPR) as $\pi_\theta$ evolves, we implement a trend-aware closed-loop controller. Let $PR_k$ be the pass rate at iteration $k$, and $\Delta_k = PR_k - \text{TPR}$. The threshold $T$ evolves via the following dynamics:

$$\begin{cases} T_{k+1} & = T_k^{1+(\mu-\epsilon_0 \cdot \mathbb{I}[\tau_k \Delta_k < 0])\Delta_k} \\ \tau_{k+1} & = \eta \tau_k + (1-\eta)(T_{k+1} - T_k) \end{cases} \quad (7)$$

where $\mu$ controls sensitivity, $\eta$ is momentum decay, and the indicator $\mathbb{I}[\tau_k \Delta_k < 0]$ reduces gain upon trend conflict. This ensures that $T$ robustly tracks the $(1-\text{TPR})$-quantile of $\tilde{R}$, filtering stochastic noise while preserving high information density in every gradient step.

## 4. Experiment

In this section, we evaluate the efficacy of Sync-R1 across a diverse range of personalized understanding and generation tasks. Our experiments are designed to investigate: (1) whether the synergistic loop improves fine-grained multi-modal alignment; (2) the efficiency gains provided by the DGS mechanism; and (3) the framework's robustness across different initialization recipes.

### 4.1. Experiment Setup

**Implementation Details.** We utilize Show-o (1.3B, 512×512) (Xie et al., 2025a) as the multi-modal backbone.

We adopt the multi-stage supervised fine-tuning protocol established in An et al. (2025b) to initialize the personalized tokens. In the RL stage, we set the group size $G = 9$ and maintain a balanced 1:1 ratio between Visual Instruct Reasoning and Text Attribute Reasoning. All models are trained on $8\times$ NVIDIA H100 GPUs. Detailed hyperparameters and SFT training protocols are provided in Appendix E.

**Baseline Comparisons.** We benchmark Sync-R1 against a comprehensive suite of models across diverse paradigms: (1) unified architectures, including UniCTokens (An et al., 2025b) (adopting its official training protocol) and Yo'Chameleon (Nguyen et al., 2025) (retrained at 7B scale with 1k images per concept); (2) specialized understanding-only and generation-only models; and (3) GPT-4o, serving as an approximate upper bound. Further implementation details are provided in Appendix E.

**Dataset.** To evaluate high-order inferential logic and dense semantic synthesis, we augment UnifyBench into Unify-Bench++. For understanding, we incorporate Reasoning and Dense Reasoning tasks to test concept disambiguation. For generation, we introduce Reasoning Generation alongside Dense Generation and Dense Personalized Generation to assess fine-grained alignment. This creates a rigorous stress-test for synergistic multi-modal reasoning (see Appendix B for details).

**Multi-modal Evaluation Suite.** For reasoning accuracy, we employ BLEU and GPT-4o-based semantic scoring to ensure precise textual understanding. For generative fidelity, we employ CLIP-T and GPT-4o to assess the visual alignment with dense prompts and inferred.

*Table 1.* **Quantitative Results on Unifybench++.** TP = Text Prompt. IP = Image Prompt. Columns with * are newly extended from Unifybench. Best and second best performances are highlighted.

| Type | Method | Model Size | Und. | | | | | | | Gen. | | | | | | | |
|---|---|---|---|---|---|---|---|---|---|---|---|---|---|---|---|---|---|
| | | | Rec. | Rea.* | Dense Rea.* | VQA | | QA | | Pure Gen. | | Dense Gen.* | | Rea. Gen.* | | Dense Rea. Gen.* | |
| | | | Weight | BLEU | GPT | BLEU | GPT | BLEU | GPT | CLIP-T | CLIP-I | GPT | CLIP-I | CLIP-T | CLIP-I | GPT | CLIP-I |
| **Upper Bound** | GPT-4o+TP | 200B | 0.742 | 0.746 | 0.819 | 0.784 | 0.863 | 0.611 | 0.699 | 0.306 | 0.684 | 0.374 | 0.697 | 0.357 | 0.776 | 0.403 | 0.692 |
| | GPT-4o+IP | 200B | 0.788 | 0.745 | 0.854 | 0.751 | 0.784 | 0.594 | 0.658 | 0.309 | 0.787 | 0.382 | 0.672 | 0.376 | 0.804 | 0.381 | 0.796 |
| | Real Images | - | - | - | - | - | - | - | - | - | 0.833 | - | - | - | - | - | - |
| **Und. Only** | Yo'LLaVA | 13B | 0.921 | 0.327 | 0.529 | 0.616 | 0.625 | 0.614 | 0.594 | - | - | - | - | - | - | - | - |
| | MC-LLaVA | 13B | 0.924 | 0.297 | 0.511 | 0.623 | 0.636 | 0.606 | 0.583 | - | - | - | - | - | - | - | - |
| | RAP-MLLM | 13B | 0.936 | 0.332 | 0.595 | 0.624 | 0.617 | 0.712 | 0.723 | - | - | - | - | - | - | - | - |
| | Qwen2.5-VL + TP | 3B | 0.669 | 0.218 | 0.341 | 0.404 | 0.726 | 0.579 | 0.770 | | - | - | - | - | - | - | - |
| | Yo'LLaVA(Phi-1.5) | 1.3B | 0.769 | 0.225 | 0.484 | 0.493 | 0.493 | 0.511 | 0.498 | - | - | - | - | - | - | - | - |
| **Gen. Only** | Text inversion | 1.0B | - | - | - | - | - | - | - | 0.248 | 0.632 | 0.322 | 0.538 | 0.294 | 0.673 | 0.351 | 0.633 |
| | DreamBooth (SD) | 1.0B | - | - | - | - | - | - | - | 0.282 | 0.645 | 0.323 | 0.558 | 0.313 | 0.661 | 0.367 | 0.651 |
| **Unified Model** | Chamaleon+TP | 7B | 0.685 | 0.206 | 0.313 | 0.411 | 0.489 | 0.509 | 0.560 | 0.186 | 0.542 | 0.281 | 0.463 | 0.193 | 0.516 | 0.309 | 0.549 |
| | Chameleon+IP | 7B | 0.497 | 0.216 | 0.374 | 0.446 | 0.497 | 0.411 | 0.532 | 0.165 | 0.514 | 0.266 | 0.449 | 0.190 | 0.532 | 0.299 | 0.503 |
| | Show-o+TP | 1.3B | 0.562 | 0.217 | 0.337 | 0.462 | 0.412 | 0.507 | 0.579 | 0.263 | 0.663 | 0.299 | 0.558 | 0.235 | 0.679 | 0.341 | 0.660 |
| | Yo'Chameleon | 7B | 0.764 | 0.231 | 0.399 | 0.470 | 0.511 | 0.506 | 0.582 | 0.235 | 0.697 | 0.289 | 0.601 | 0.273 | 0.704 | 0.321 | 0.702 |
| | UniCTokens | 1.3B | 0.792 | 0.238 | 0.385 | 0.505 | 0.521 | 0.546 | 0.601 | 0.280 | 0.750 | 0.298 | 0.639 | 0.282 | 0.762 | 0.317 | 0.712 |
| | Sync-R1 | 1.3B | 0.859 | 0.250 | 0.503 | 0.592 | 0.606 | 0.604 | 0.652 | 0.308 | 0.765 | 0.337 | 0.645 | 0.324 | 0.801 | 0.353 | 0.756 |

## 4.2. Synergistic Reward Ensemble

To close the feedback loop between discrete reasoning and generation, we define a unified reward function:

$$R_{total} = w_1 R_{TIER} + w_2 R_{BER} + w_3 R_{DER} + w_4 R_{FER} \quad (8)$$

The ensemble incorporates: (1) TIER (Text Inference Evaluation) via ERNIE 3.0 (Sun et al., 2021) to measure logical consistency; (2) BER (BLIP Evaluation (Li et al., 2023)) for cross-modal semantic alignment; (3) DER (DINOv2 Evaluation (Oquab et al., 2024)) for structural identity preservation of personalized concepts; and (4) FER (Facenet Evaluation) for high-fidelity facial synthesis. Scaling coefficients $w_i$ are calibrated via sensitivity analysis (see Appendix G).

## 4.3. Main Results

**Performance on UnifyBench.** We first evaluate the foundational capabilities of Sync-R1 on standard personalized understanding tasks, including Recognition (Rec.), Visual Question Answering (VQA), and Question Answering (QA) (Alaluf et al., 2024). As presented in Table 1, despite possessing only 1.3B parameters, Sync-R1 delivers exceptional performance, achieving an average improvement of **12.2%** over previous state-of-the-art (SOTA) methods across all understanding metrics. In terms of generation, we utilize the Pure Gen. metric to assess identity-preserving generation capabilities. Sync-R1 consistently outperforms other unified models, securing the top rank. Collectively, these results corroborate that our framework offers a highly resource-efficient solution for unifying personalized understanding and generation without compromising performance on established benchmarks.

**Performance on UnifyBench++.** The primary contribution of our evaluation lies in UnifyBench++, which is designed to stress-test the model's capacity for reasoning-intensive and information-dense scenarios. As shown in Table 1, Sync-R1 achieves SOTA results across all newly introduced metrics, validating the effectiveness of our Sync-GRPO strategy. **(1) Understanding Reasoning:** We observe substantial gains in the Dense Rea. metric compared to baselines. This strongly indicates that our model has transcended simple pattern matching and acquired a deeper, more granular understanding of personalized concepts embedded within complex contexts. **(2) Generation Reasoning:** In generation tasks requiring reasoning and high information density, our method demonstrates a significant leap over the baseline UniCTokens. Specifically, we achieve relative improvements of **14.1%** in Dense Gen., **10.0%** in Rea. Gen., and **8.8%** in Dense Rea. Gen. **(3) Qualitative Analysis:** As visualized in Figure 4, we evaluate fidelity across a complexity spectrum, ranging from Pure Generation to challenging Dense Reasoning Generation. Crucially, Sync-R1 demonstrates superior robustness in both explicit structural arrangement and implicit attribute inference. In Dense Generation, it maintains precise spatial relations under heavy semantic load (e.g., correctly positioning $\langle em \rangle$ *beside* the lamppost), whereas baselines often suffer from spatial dislocation. Furthermore, in Reasoning scenarios, Sync-R1 successfully retrieves and visualizes implicit attributes (e.g., the painting of flowers) where previous methods succumb

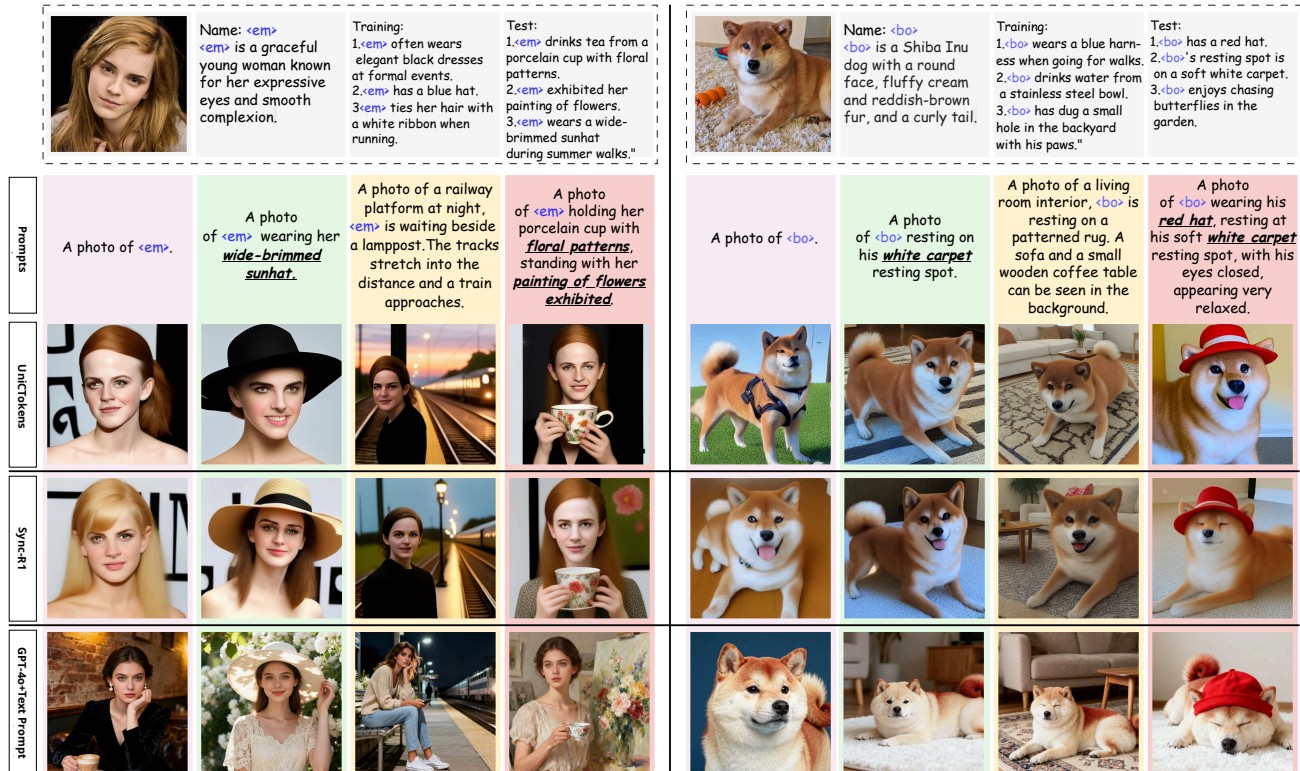

Figure 4. **Qualitative Comparison.** We present a visual comparison between Sync-R1 and baseline methods. The underlined text highlights specific, detailed information that requires the model to infer from the personalized context correctly.

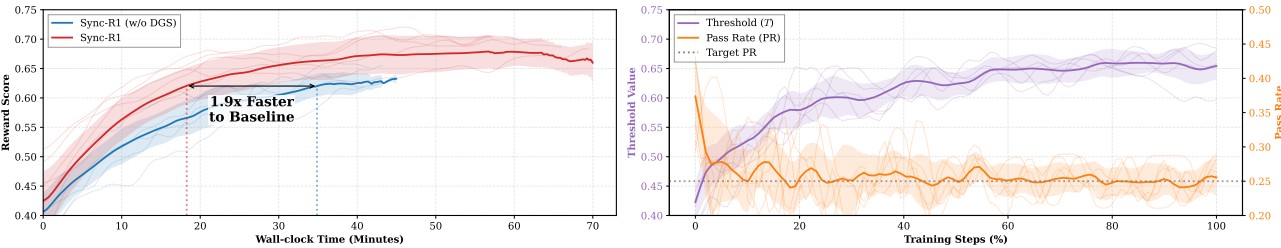

Figure 5. **Efficiency and Dynamics of DGS. (a) Convergence Efficiency:** Sync-R1 (Red) reaches 98% of the baseline's peak performance approx. **1.9× faster** in wall-clock time, confirming that high sample efficiency outweighs the selection overhead. **(b) Adaptive Dynamics:** The threshold $T$ (Purple) dynamically adapts to pass rate fluctuations (Orange) to anchor the selection ratio at TPR, ensuring robust training stability across diverse concepts.

to catastrophic neglect. This confirms that our framework enhances both the spatial coherence of complex scenes and the semantic depth of personalized understanding.

### 4.4. Ablation Study

In this section, we dissect the contribution of each component within Sync-R1, focusing on the necessity of the explicit synergistic loop and the efficiency gains from Dynamic Group Scaling (DGS).

**Efficacy of the Explicit Synergistic Loop.** To validate the hypothesis that personalized understanding and generation

mutually enhance one another, we conduct a component-wise ablation by selectively disabling parts of the reasoning trajectory (Table 2). The removal of Visual Instruct Reasoning leads to a precipitous drop in MMU metrics (e.g., Rec.) and degrades generative fidelity, confirming that explicit extraction of visual attributes is essential for semantic guidance. Similarly, excluding Textual Attribute Reasoning causes a severe regression in logic-intensive metrics (e.g., Rea. and Rea. Gen.), establishing this module as the primary driver for resolving semantic ambiguities. Most critically, when the entire understanding loop is removed—reducing the method to standard RL-tuned gener-

*Table 2.* **Ablation study on the effectiveness of the two reasoning part and DGS method.**

| Method | Und. | | | | | | | Gen. | | | | | | | |
|---|---|---|---|---|---|---|---|---|---|---|---|---|---|---|---|
| | Rec. | Rea.* | Dense Rea.* | VQA | | QA | | Pure Gen. | | Dense Gen.* | | Rea. Gen.* | | Dense Rea. Gen.* | |
| | Weight | BLEU | GPT | BLEU | GPT | BLEU | GPT | CLIP-T | CLIP-I | GPT | CLIP-I | CLIP-T | CLIP-I | GPT | CLIP-I |
| w/o Und.Rea. | 0.808 | 0.224 | 0.427 | 0.511 | 0.519 | 0.549 | 0.621 | 0.301 | 0.762 | 0.323 | 0.637 | 0.289 | 0.765 | 0.321 | 0.727 |
| w/o VIA | 0.823 | 0.239 | 0.489 | 0.587 | 0.591 | 0.571 | 0.626 | 0.299 | 0.759 | 0.306 | 0.641 | 0.301 | 0.799 | 0.343 | 0.741 |
| w/o TIA | 0.855 | 0.222 | 0.431 | 0.572 | 0.588 | 0.597 | 0.646 | 0.305 | 0.757 | 0.319 | 0.639 | 0.291 | 0.773 | 0.329 | 0.728 |
| w/o DSOG | 0.848 | 0.243 | 0.498 | 0.590 | 0.583 | 0.607 | 0.643 | 0.305 | 0.766 | 0.331 | 0.632 | 0.307 | 0.796 | 0.338 | 0.739 |
| Sync-R1 | 0.859 | 0.250 | 0.503 | 0.592 | 0.606 | 0.604 | 0.652 | 0.308 | 0.765 | 0.337 | 0.645 | 0.324 | 0.801 | 0.353 | 0.756 |

*Table 3.* **Ablation study on different initial methods.**

| Method | Und. | | | | | | | Gen. | | | | | | | |
|---|---|---|---|---|---|---|---|---|---|---|---|---|---|---|---|
| | Rec. | Rea.* | Dense Rea.* | VQA | | QA | | Pure Gen. | | Dense Gen.* | | Rea. Gen.* | | Dense Rea. Gen.* | |
| | Weight | BLEU | GPT | BLEU | GPT | BLEU | GPT | CLIP-T | CLIP-I | GPT | CLIP-I | CLIP-T | CLIP-I | GPT | CLIP-I |
| Joint | 0.712 | 0.224 | 0.387 | 0.489 | 0.482 | 0.529 | 0.577 | 0.264 | 0.680 | 0.261 | 0.607 | 0.266 | 0.724 | 0.288 | 0.653 |
| Joint+Sync-GRPO | 0.789 | 0.231 | 0.468 | 0.548 | 0.557 | 0.581 | 0.629 | 0.287 | 0.695 | 0.291 | 0.616 | 0.303 | 0.757 | 0.322 | 0.689 |
| UniCTokens | 0.792 | 0.238 | 0.385 | 0.505 | 0.521 | 0.546 | 0.601 | 0.280 | 0.750 | 0.298 | 0.639 | 0.282 | 0.762 | 0.317 | 0.712 |
| Sync-R1 | 0.859 | 0.250 | 0.503 | 0.592 | 0.606 | 0.604 | 0.652 | 0.308 | 0.765 | 0.337 | 0.645 | 0.324 | 0.801 | 0.353 | 0.756 |

ation—we observe a systemic collapse across all personalization metrics. While standard aesthetic scores (Pure Gen.) remain stable, the model loses its ability to align with user-specific contexts (e.g., Rea. Gen.). This decoupling confirms our core premise: generation quality alone does not equate to personalization; explicit understanding is the prerequisite for faithful user alignment.

**Efficiency and Dynamics of DGS.** We analyze the training dynamics in Figure 5. As illustrated in Figure 5(a), DGS achieves a $1.9\times$ wall-clock acceleration, confirming that the benefits of gradient variance reduction significantly outweigh the selection overhead. The underlying mechanism is visualized in Figure 5(b), where the threshold (Purple) automatically trends upward in correlation with the model's increasing competence. This acts as a homeostatic regulator, anchoring the pass rate (Orange) to the target ratio. By establishing this implicit self-pacing curriculum, DGS ensures robust stability and continuous learning from high-value trajectories across diverse concepts without manual hyperparameter tuning.

**Robustness to Initialization Strategies.** Finally, we investigate whether Sync-R1 relies on complex cold-start procedures (e.g., An et al. (2025b)). Results in Table 3 indicate that Sync-R1 achieves consistent SOTA performance regardless of initialization. Notably, applying Sync-GRPO to a simple "Joint Training" baseline yields a 17.9% improvement in Dense Rea. and an average increase of 13.8% in VQA tasks, surpassing previous complex SOTA. The

method also achieves a 7.5% average gain in generation metrics, with CLIP-T scores even exceeding those of UniCTokens. This proves that Sync-R1 serves as a robust, universal post-training paradigm that democratizes high-performance personalization without demanding elaborate pre-training recipes.

## 5. Conclusion

We present **Sync-R1**, a unified framework that orchestrates the synergistic co-optimization of personalized understanding and generation. By establishing an explicit reasoning loop via **Sync-GRPO**, we bridge the gap between comprehension and creation, enabling these tasks to mutually reinforce one another within a closed-loop reinforcement learning process. This approach empowers the model to transcend simple pattern matching, achieving deeper reasoning capabilities essential for interpreting dense user contexts and generating high-fidelity personalized content. Empirical results on **UnifyBench++** demonstrate that Sync-R1 significantly outperforms existing baselines, particularly in reasoning-intensive and information-dense scenarios. Furthermore, our **Dynamic Group Scaling (DGS)** strategy addresses the computational bottlenecks of multimodal RL, accelerating convergence by filtering high-potential trajectories. In summary, this work establishes a new paradigm for leveraging reinforcement learning to align multimodal understanding and generation, paving the way for more robust, efficient, and reasoning-driven personalization systems.

## Impact Statement

This paper presents work whose goal is to advance the field of Machine Learning. There are many potential societal consequences of our work, none of which we feel must be specifically highlighted here.

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

## A. Other Related Work

**Reinforcement Learning in Large Models.** Within the LLM domain, recent studies (Chu et al., 2025) have made significant strides in addressing critical challenges such as long-chain reasoning, output coherence, and training efficiency. Following the introduction of the GRPO strategy and the demonstration of rule-based rewards in DeepSeek-R1 (DeepSeek-AI et al., 2025), there has been a surge of research applying these principles to Multimodal LLMs (MLLMs) (Huang et al., 2025; Pan et al., 2025a; Zhou et al., 2025). This expansion spans diverse applications, including semantic segmentation (Liu et al., 2025a), object recognition (Liu et al., 2025d), video analysis (Zhang et al., 2025a; Sun et al., 2025), code generation (Austin et al., 2021; Jain et al., 2024), and mathematical problem-solving (Hendrycks et al., 2021; Zhang et al., 2024a;b). Ample evidence suggests that RL not only enhances in-domain reasoning capabilities but also yields superior performance compared to Supervised Fine-Tuning (SFT) in out-of-distribution (OOD) scenarios.

**Personalized Models.** The core objective of model personalization is to precisely integrate concept-specific information into model outputs via diverse techniques. In the generative domain, recent approaches focus on recontextualization conditioned on text. For instance, DreamBooth (Ruiz et al., 2023) ensures subject authenticity in generated results, while Textual Inversion (Gal et al., 2022) optimizes special tokens using soft-prompt tuning. In the realm of LLMs, personalization has been achieved through dual-tower structures that endow models with user-aware capabilities (Pi et al., 2024). For MLLMs, recent works (Alaluf et al., 2024; Hao et al., 2025; An et al., 2025a) enhance output relevance by organically combining user data with visual content via fine-tuning or Retrieval-Augmented Generation (RAG). Within Unified MLLMs, Chameleon (Team, 2025) employs "text embedding optimization + Transformer fine-tuning" for information injection, while Yo'Chameleon (Nguyen et al., 2025) utilizes soft prompts. UniCTokens (An et al., 2025a) was the pioneer in focusing on the mutual promotion between understanding and generation, significantly reducing sample requirements and expanding the scope of personalization. However, despite these advancements, existing unified methods often suffer from cumbersome training protocols and suboptimal performance in complex semantic generation tasks, highlighting an urgent need for more efficient and robust solutions.

**Unifying Understanding and Generation.** Ideally, a single model should seamlessly handle both understanding and generation tasks. To this end, numerous studies have explored unified architectures. For example, Ge et al. (2023); Dong et al. (2024); Ge et al. (2025) aggregate language-conditioned information to drive generation, while Janus (Wu et al., 2024a) models different modalities using distinct tokenizers. Show-o (Xie et al., 2025a) and Transfusion (Zhou et al., 2024) combine autoregressive and diffusion methods to process text and images, whereas Emu3 (Wang et al., 2024) and Selftok (Wang et al., 2025) unify multimodal data into discrete tokens for joint training. In personalization scenarios, the cross-modal potential of unified models inspires the pursuit of complementary advantages. While early attempts serially combined understanding and generation models (Luo et al., 2024; Dunlap et al., 2023; Liu et al., 2025c), they faced challenges in end-to-end optimization. Subsequent efforts in joint training with shared representations (Fang et al., 2023; Tong et al., 2024; Fan et al., 2024) achieved co-optimization but often lacked deep synergy. In this work, we leverage reinforcement learning to amplify the mutual promotion between generation and understanding, enhancing the model's reasoning ability regarding conceptual information and significantly improving performance in complex, reasoning-intensive generation tasks.

## B. Details of Dataset

Building upon the foundation of UnifyBench (An et al., 2025a), we introduce **UnifyBench++**, an augmented benchmark designed to rigorously evaluate advanced reasoning and generation capabilities. As illustrated in Figure 6, we enriched the dataset structure by associating each concept with distinct pieces of supplementary context (denoted as `extra_info`), each paired with a specific reasoning prompt. The italicized segments in the figure represent the training data—comprising the `extra_info` and reasoning prompts exposed to Sync-R1 during the optimization phase. Conversely, the non-italicized segments are strictly held out for evaluation under the Reasoning (`Rea.`) and Reasoning Generation (`Rea. Gen.`) protocols. Furthermore, to assess robustness in high-complexity settings, we constructed three specialized subsets. Dense Generation (`Dense Gen.`) targets the generative domain, challenging the model to maintain fidelity under high information load. Dense Reasoning (`Dense Rea.`) focuses on the understanding domain, evaluating fine-grained comprehension capabilities in complex scenarios. Crucially, Dense Reasoning Generation (`Dense Rea. Gen.`) bridges these modalities by assessing the model's ability to synthesize visual content contingent upon attributes inferred from the understanding process.

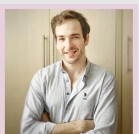 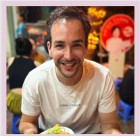

**name:** ‹w_n›
**info:** ‹w_n› is a cheerful young man with short wavy brown hair and fair skin.

extra_info:
*1.‹w_n› has a blue T-shirt.*
*2.‹w_n› enjoys eating street food while casually dressed.*
*3.‹w_n› wears his green athletic short-sleeved shirt when he goes running.*
*4.‹w_n› will wear an afro-shaped wig at the party.*
*5.‹w_n› works in a biological laboratory.*

Rea. Gen.:
*1.A photo of ‹w_n› wearing his T-shirt.*
*2.A photo of ‹w_n› eating street food.*
*3.A photo of ‹w_n› running.*
4.A photo of ‹w_n› wearing his wig at a party.
5.A photo of ‹w_n› working.

**Dense:** 1.A photo of a lively street in Hanoi, ‹w_n› is walking past food stalls. Red lanterns hang overhead and scooters are parked along the road. 2.A photo of ‹w_n› sitting on a wooden boat in a river. Green hills rise in the distance and water ripples around the boat. 3.A photo of a bright café interior, ‹w_n› is leaning on a counter. Hanging plants and chalkboard menus decorate the wall.

**Dense Rea:** A photo of ‹w_n› working, dressing in a white lab coat, holding a test tube in his hand. He appears to be looking anxious.

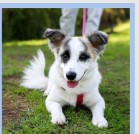 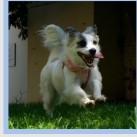

**name:** ‹m›
**info:** ‹m› is a small white dog with fluffy fur, a bushy tail, and distinctive dark markings around the eyes and ears.

extra_info:
*1.‹m› wears a pink bib when going out.*
*2.‹m› always rests on a sofa when indoors.*
*3.‹m› owns a plush toy shaped like a rabbit.*
*4.‹m› enjoys digging in the sand at the beach.*
*5.‹m› went on an adventure in the forest on last year's birthday.*

Rea. Gen.:
*1.A photo of ‹m› going out wearing its bib.*
*2.A photo of ‹m› resting indoors.*
*3.A photo of ‹m› with her plush toy.*
4.A photo of ‹m› doing what she loves at the beach.
5.A photo of ‹m› on an adventure during last year's birthday.

**Dense:** 1.A photo of a quiet suburban street, ‹m› is standing on the sidewalk. A row of houses and parked bicycles can be seen in the background. 2.A photo of ‹m› sitting on a wooden bench in a park. Tall trees and a walking path stretch behind the bench. 3.A photo of a sunny kitchen interior, ‹m› is lying on the tiled floor. A dining table and chairs are placed nearby.

**Dense Rea:** A photo of ‹m› doing what she loves at the beach, with a sandcastle nearby. The background shows the ocean and a clear sky.

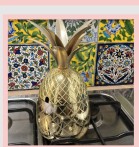 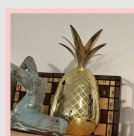

**name:** ‹g_p›
**info:** ‹g_p› is a glossy golden pineapple-shaped container with a crisscross texture and spiky metallic leaves.

extra_info:
*1.The price of ‹g_p› is $50.*
*2.The interior of the ‹g_p› is hollow.*
*3.‹g_p› is placed on a wooden table during holidays.*

Rea. Gen.:
*1.A photo of a ‹g_p› displayed on a sales shelf, with the price clearly marked beside it.*
*2.A photo of ‹g_p› reveals its interior.*
3.A photo of ‹g_p› during holidays.

**Dense:** 1.A photo of a modern living room, ‹g_p› is placed on a glass coffee table. A gray sofa and a floor lamp stand behind it. 2.A photo of ‹g_p› sitting on a marble countertop in a kitchen. A bowl of fruit and a hanging light fixture are nearby. 3.A photo of an outdoor patio at sunset, ‹g_p› rests on a wooden table. String lights hang overhead and potted plants surround the space.

**Dense Rea:** A photo of ‹g_p› during holidays, with festive decorations around it, and warm lighting.

*Figure 6.* **Snapshot of UnifyBench++.** We present illustrative examples of concept entries within our constructed dataset. The *italicized segments* within the `extra_info` and `Rea. Gen.` columns denote the specific data subsets utilized for training Sync-R1, while the non-italicized text is reserved for evaluation.

## C. Theoretical Formulation of Sync-GRPO

**Foundational Strategy.** We build our optimization framework upon the **Dr.GRPO** strategy (Liu et al., 2025b), which mitigates critical optimization biases inherent in standard GRPO. Dr.GRPO fundamentally diverges from the standard paradigm through two synergistic modifications: First, it discards the variance normalization term ($\sigma$). Instead of $\hat{A}_i = (R_i - \mu)/\sigma$, it computes advantage via strict group centering: $\hat{A}_i = R_i - \frac{1}{G}\sum_{k=1}^{G} R_k$. This preserves the absolute magnitude of reward differences, ensuring gradients accurately reflect semantic difficulty and preventing noise amplification in low-variance groups. Besides, it eliminates the token-level length normalization ($1/|o_i|$) from the objective. By treating the generation trajectory as a holistic unit (via summation $\sum_{j=1}^{|o_i|}$ rather than averaging), Dr.GRPO effectively curtails the model's tendency to exploit reward structures through verbosity, ensuring concise alignment with prompt constraints.

**The Optimization Objective of Sync-GRPO.**    To adapt this foundation to our unified multi-modal setting—where the model must simultaneously perform intermediate reasoning (Understanding) and visual synthesis (Generation)—we propose **Sync-GRPO**. This formulation introduces a dynamic weighting mechanism to balance the distinct gradient contributions of the understanding and generation phases. The final optimization objective is formulated as:

$$\mathcal{J}_{\text{Sync-GRPO}}(\theta) = \mathbb{E}_{(q,a)\sim\mathcal{D},\{o_i\}_{i=1}^{G}\sim\pi_{\theta_{\text{old}}}(\cdot|q)}$$
$$\left[ \frac{1}{G} \sum_{i=1}^{G} \sum_{j=1}^{|o_i|} \alpha_j \left( \min\left( D_{i,j}(\theta)\hat{A}_i, \text{clip}\left( D_{i,j}(\theta), 1-\varepsilon, 1+\varepsilon \right)\hat{A}_i \right) - \beta \mathbb{D}_{\text{KL}}(\pi_\theta \parallel \pi_{\text{ref}}) \right) \right] \tag{9}$$

Here, the task-specific weighting coefficient $\alpha_j$ is defined piecewise to modulate the learning focus across the sequence:

$$\alpha_j = \begin{cases} \alpha_{\text{text}}, & j \in \text{Tokens in } IR_i \\ \alpha_{\text{image}}, & j \in \text{Image Tokens} \end{cases} \tag{10}$$

where $IR_i$ denotes the length of the intermediate reasoning segment (Understanding Task), and the subsequent tokens correspond to the generation phase. By tuning $\alpha_{\text{text}}$ and $\alpha_{\text{image}}$, Sync-GRPO ensures robust joint optimization, preventing one modality from dominating the learning process while maintaining the rigorous constraints of Dr.GRPO.

## D. Theoretical Analysis and Proofs

In this section, we provide the detailed mathematical derivations supporting the effectiveness of Sync-GRPO and Dynamic Group Scaling (DGS). We proceed in a bottom-up manner: first deriving the gradient variance, then proving the variance reduction and SNR amplification properties of DGS, and finally establishing the control stability.

### D.1. Gradient Derivation and Variance Decomposition

To justify the focus of DGS on minimizing reward variance, we first explicitly derive the analytical form of the gradient estimator $\hat{g}$ under the clipping mechanism and then decompose its variance.

**Step 1: Derivation of the Gradient Estimator $\hat{g}$.**    Recall the sample-based objective function for Sync-GRPO:

$$\hat{\mathcal{J}}(\theta) = \frac{1}{BG} \sum_{b,i,j} \alpha_j \underbrace{\min\left( D_{i,j}\hat{A}_i, \text{clip}(D_{i,j}, 1-\varepsilon, 1+\varepsilon)\hat{A}_i \right)}_{\mathcal{L}_{i,j}^{\text{CLIP}}(\theta)} - \beta \mathbb{D}_{\text{KL}} \tag{11}$$

We compute the gradient $\nabla_\theta \mathcal{L}_{i,j}^{\text{CLIP}}(\theta)$ by analyzing the behavior of the $\min$ operator. The clipping function restricts $D_{i,j}$ to the interval $[1-\varepsilon, 1+\varepsilon]$. Differentiating with respect to $\theta$:

- **Case 1: Active Region.** The probability ratio is within the trust region, or the update moves $D_{i,j}$ towards the region. In this case, $\min(\cdot) = D_{i,j}\hat{A}_i$. Using the log-derivative trick $\nabla_\theta D_{i,j} = D_{i,j}\nabla_\theta \log \pi_\theta(o_{i,j})$, the gradient is:

$$\nabla_\theta(D_{i,j}\hat{A}_i) = \hat{A}_i \cdot D_{i,j}\nabla_\theta \log \pi_\theta(o_{i,j})$$

- **Case 2: Clipped Region.** The term is clipped to $(1 \pm \varepsilon)\hat{A}_i$. Since $(1 \pm \varepsilon)\hat{A}_i$ is a constant with respect to the current policy parameters $\theta$ (locally), its gradient is zero:

$$\nabla_\theta((1 \pm \varepsilon)\hat{A}_i) = 0$$

Let $\mathbb{I}_{i,j}^{\text{active}}$ be the indicator function for the active region. The total policy gradient estimator $G_{\text{PG}}$ is:

$$G_{\text{PG}} = \frac{1}{BG} \sum_{b,i,j} \mathbb{I}_{i,j}^{\text{active}} \cdot \alpha_j D_{i,j}\hat{A}_i \nabla_\theta \log \pi_\theta(o_{i,j}) \tag{12}$$

**Step 2: Variance Decomposition.** Based on Step 1, the total gradient variance is:

$$\text{Var}(\hat{g}) = \text{Var}(G_{\text{PG}} - \beta\nabla\mathbb{D}_{\text{KL}}) \tag{13}$$

Since the KL-divergence term depends only on the policy distribution (which is deterministic given $\theta$) and does not involve the stochastic sampling of rewards, its variance is negligible compared to the Monte Carlo estimator $G_{\text{PG}}$. Thus, we focus on analyzing $\text{Var}(G_{\text{PG}})$.

**Step 3: Variance Analysis of the Policy Gradient Term.** Consistent with the *linearized policy gradient* regime assumed in Theorem 3.1 of the main text, we omit the clipping mechanism (i.e., setting $\mathbb{I}_{i,j}^{\text{active}} = 1$) for this variance analysis. Let the unclipped gradient contribution of a single sample be:

$$\mathbf{x}_{b,i,j} = \underbrace{(\alpha_j D_{i,j}\nabla_\theta\log\pi_\theta(o_{i,j}))}_{\mathbf{u}_{b,i,j}} \cdot \hat{A}_i \tag{14}$$

We assume that the fluctuations in $\mathbf{u}_{b,i,j}$ are statistically independent of the fluctuations in the advantage estimate $\hat{A}_i$.

Applying the variance of a product for independent variables, and noting that the advantage function is typically centered (i.e., $\mathbb{E}[\hat{A}_i] \approx 0$ due to the baseline), the variance simplifies as follows:

$$\text{Var}(\mathbf{x}_{b,i,j}) \approx \mathbb{E}[\|\mathbf{u}_{b,i,j}\|^2] \cdot \text{Var}(\hat{A}_i) \tag{15}$$

This indicates that the variance of the gradient update is the product of the expected squared norm of $\mathbf{u}_{b,i,j}$ and the variance of the advantage. Consequently, the trace of the variance for the aggregated estimator is:

$$\text{Tr}(\text{Var}[G_{\text{PG}}]) = \frac{1}{(BG)^2}\sum_{b,i,j}\mathbb{E}[\|\mathbf{u}_{b,i,j}\|^2] \cdot \text{Var}(\hat{A}_i) \tag{16}$$

**Conclusion.** Since $\hat{A}_i = R_i - \text{mean}\left(\{R_i\}_{i=1}^G\right)$ and the baseline mean $\left(\{R_i\}_{i=1}^G\right)$ is deterministic within the group, we have $\text{Var}(\hat{A}_i) = \text{Var}(R_i)$. Based on the assumption in the main text that the rewards $R$ are independent and identically distributed (i.i.d.), we obtain:

$$\text{Tr}(\text{Var}[\hat{g}]) \propto \text{Var}[\hat{A}] \tag{17}$$

This derivation formally proves that reducing the variance of the reward distribution (via DGS) directly minimizes the variance of the optimization update.

**D.2. Proof of Variance Reduction (Theorem 3.1)**

**Assumption.** Let the terminal reward $R$ and surrogate $\tilde{R}$ follow a bivariate normal distribution:

$$\begin{pmatrix} R \\ \tilde{R} \end{pmatrix} \sim \mathcal{N}\left(\begin{pmatrix} \mu_R \\ \mu_{\tilde{R}} \end{pmatrix}, \begin{pmatrix} \sigma_R^2 & \rho\sigma_R\sigma_{\tilde{R}} \\ \rho\sigma_R\sigma_{\tilde{R}} & \sigma_{\tilde{R}}^2 \end{pmatrix}\right) \tag{18}$$

with $\rho > 0$. Let $\mathcal{E} = \{\tilde{R} > T\}$ be the selection event with $P(\mathcal{E}) = G/N$.

**Proof.** We verify the variance of the selected rewards using the Law of Total Variance:

$$\text{Var}[R \mid \mathcal{E}] = \mathbb{E}[\text{Var}(R \mid \tilde{R}) \mid \mathcal{E}] + \text{Var}(\mathbb{E}[R \mid \tilde{R}] \mid \mathcal{E}) \tag{19}$$

The conditional variance $\text{Var}(R \mid \tilde{R}) = (1 - \rho^2)\sigma_R^2$ is constant. Thus, the first term $\mathbb{E}[\text{Var}(R \mid \tilde{R}) \mid \mathcal{E}] = (1 - \rho^2)\sigma_R^2$.

For the second term, the conditional expectation is linear: $\mathbb{E}[R \mid \tilde{R}] = \mu_R + \rho\frac{\sigma_R}{\sigma_{\tilde{R}}}(\tilde{R} - \mu_{\tilde{R}})$. Let $Y$ be the random variable $\tilde{R}$ conditioned on $\tilde{R} > T$ (a truncated normal distribution). The variance of this term is:

$$\text{Var}(\mathbb{E}[R \mid \tilde{R}] \mid \mathcal{E}) = \left(\rho\frac{\sigma_R}{\sigma_{\tilde{R}}}\right)^2\text{Var}(Y) \tag{20}$$

For a standard normal distribution, the variance of the truncated top $p$-quantile satisfies $Var(Z|Z > z_p) \leq p$ for $p \leq 0.5$. This follows from the fact that the conditional variance increases with $p$ and equals $p$ when $p = 1$. More precisely, for a normal variable $\tilde{R}$ with mean $\mu_{\tilde{R}}$ and variance $\sigma_{\tilde{R}}^2$, we have:

$$\text{Var}(Y) = \sigma_{\tilde{R}}^2 \cdot \text{Var}\left(\frac{\tilde{R} - \mu_{\tilde{R}}}{\sigma_{\tilde{R}}} \,\middle|\, \frac{\tilde{R} - \mu_{\tilde{R}}}{\sigma_{\tilde{R}}} > \frac{T - \mu_{\tilde{R}}}{\sigma_{\tilde{R}}}\right) \leq \sigma_{\tilde{R}}^2 \cdot \frac{G}{N} \tag{21}$$

where the inequality holds when $G/N \leq 0.5$ (which is satisfied in our setting with $\frac{G}{N} \approx TPR = 0.25 < \frac{1}{2}$). Specifically, $\text{Var}(Y) \leq \frac{G}{N}\sigma_{\tilde{R}}^2$. Substituting this bound:

$$\text{Var}(\mathbb{E}[R \mid \tilde{R}] \mid \mathcal{E}) \leq \rho^2 \frac{\sigma_R^2}{\sigma_{\tilde{R}}^2} \cdot \frac{G}{N}\sigma_{\tilde{R}}^2 = \rho^2 \sigma_R^2 \frac{G}{N} \tag{22}$$

*Synthesis:*

$$\text{Var}[R \mid \mathcal{E}] \leq (1 - \rho^2)\sigma_R^2 + \rho^2 \frac{G}{N}\sigma_R^2 \tag{23}$$

$$= \sigma_R^2\left(1 - \rho^2\left(1 - \frac{G}{N}\right)\right) \tag{24}$$

This proves the inequality in Theorem 3.1. $\qquad\square$

### D.3. Proof of SNR Amplification (Corollary 3.2)

The Signal-to-Noise Ratio (SNR) of the gradient estimator is $\text{SNR} = \frac{\|\mathbb{E}[\hat{g}]\|^2}{\text{Tr}(\text{Var}[\hat{g}])}$.

**Numerator (Signal):** The expected gradient aligns with the expected reward of the selected samples. Under the bivariate normal assumption, $\mathbb{E}[R \mid \mathcal{E}]$ can be expressed exactly using the **Inverse Mills Ratio** $\lambda(\alpha) = \frac{\phi(\alpha)}{1 - \Phi(\alpha)} > 0$:

$$\mathbb{E}[R \mid \mathcal{E}] = \mu_R + \rho\sigma_R\lambda\left(\frac{T - \mu_{\tilde{R}}}{\sigma_{\tilde{R}}}\right) \tag{25}$$

Since $\lambda(\cdot)$ is strictly positive, $\mathbb{E}[R \mid \mathcal{E}] > \mu_R$. Thus, the signal magnitude $\|\mathbb{E}[\hat{g}_{\text{DGS}}]\|^2$ is strictly larger than the baseline $\|\mathbb{E}[\hat{g}_{\text{Base}}]\|^2$.

**Denominator (Noise):** From Section D.2, the variance is reduced by a factor $\gamma = 1 - \rho^2(1 - G/N) < 1$.

**Result:**

$$\text{SNR}_{\text{DGS}} \propto \frac{(\mu_R + \delta)^2}{\gamma\sigma_R^2} > \frac{\mu_R^2}{\sigma_R^2} \cdot \frac{1}{\gamma} = \frac{1}{1 - \rho^2(1 - \frac{G}{N})}\text{SNR}_{\text{Base}} \tag{26}$$

This confirms significant SNR amplification. $\qquad\square$

### D.4. Stability of Adaptive Threshold Control

We analyze the stability of the trend-aware controller defined in Eq. 7. Let $F_{\tilde{R}}(t)$ denote the cumulative distribution function (CDF) of the surrogate reward $\tilde{R}$ at iteration $k$. The pass rate is $PR_k = 1 - F_{\tilde{R}}(T_k)$. Our goal is for $T_k$ to converge to the value $T^*$ satisfying $PR(T^*) = \text{TPR}$.

**Linearized Dynamics.** Assuming $F_{\tilde{R}}$ is differentiable with density $f_{\tilde{R}}$, we can linearize around the equilibrium $T^*$. For small deviations $\delta_k = T_k - T^*$, we have:

$$PR_k \approx \text{TPR} - f_{\tilde{R}}(T^*)\delta_k + \mathcal{O}(\delta_k^2) \tag{27}$$

Thus $\Delta_k = PR_k - \text{TPR} \approx -f_{\tilde{R}}(T^*)\delta_k$.

Taking logarithms of the update rule (ignoring the damping term for simplicity):

$$\ln T_{k+1} = (1 + \mu\Delta_k)\ln T_k \tag{28}$$

Define $x_k = \ln T_k - \ln T^*$. Linearizing yields:

$$x_{k+1} \approx x_k - \mu f_{\tilde{R}}(T^*)T^* x_k = (1 - \mu f_{\tilde{R}}(T^*)T^*)x_k \tag{29}$$

The system is locally stable when $|1 - \mu f_{\tilde{R}}(T^*)T^*| < 1$, i.e., $0 < \mu < \frac{2}{f_{\tilde{R}}(T^*)T^*}$.

**Role of Trend-Aware Damping.** The full update includes the indicator $\mathbb{I}[\tau_k \Delta_k < 0]$, which reduces the effective gain when the momentum $\tau_k$ conflicts with the error direction $\Delta_k$. This adaptive gain scheduling prevents oscillatory divergence when the gradient direction reverses due to batch noise. The momentum term $\tau_k$ itself acts as a low-pass filter on threshold changes, smoothing transient fluctuations.

**Global Behavior.** While the linear analysis guarantees local stability, the controller's global performance relies on the quasi-stationarity of the reward distribution $F_{\tilde{R}}$. As the policy $\pi_\theta$ improves, the distribution shifts slowly relative to the controller's adaptation rate. The chosen parameters $(\mu, \eta, \epsilon_0)$ ensure that the threshold $T_k$ tracks the moving $(1 - \text{TPR})$-quantile, maintaining a consistent flow of high-quality gradients throughout training.

## E. Experimental Implementation and Additional Analysis

**Hyperparameters and Training Details.** We first complete the first two-stage training of UniCTokens (An et al., 2025b), setting the number of learnable tokens as $K = 16$ and $M = 8$, and training for 15 epochs in each stage. To manage memory efficiency, the batch size is configured to 4 for understanding tasks and 1 for generation tasks. In the subsequent Sync-GRPO stage, we set the group size to $G = 9$ and maintain a balanced 1:1 mixing ratio between Visual Instruction Reasoning and Textual Attribute Reasoning. Detailed hyperparameters are summarized in Table 4.

*Table 4.* Sync-R1 training hyperparameters.

| Name | Value | Name | Value |
|---|---|---|---|
| Group Size $G$ | 9 | Learning rate $lr$ | 1e-6 |
| $\beta$ | 0.01 | Classifier-Free Guidance Scale | 5 |
| Batchsize | 1 | Max Gradient Norm | 1.0 |
| Training Steps | 100 | Image Resolution $h \times w$ | $512 \times 512$ |
| $\alpha_{text}$ | 0.4 | $\alpha_{image}$ | 0.6 |
| $TPR$ | 0.25 | $\mu$ | 0.12 |
| $\eta$ | 0.8 | $\varepsilon_0$ | 2 |

**Baselines.** To strictly evaluate the efficacy of our proposed framework, we compare against a diverse set of state-of-the-art baselines, including our direct predecessor, unified systems, and specialized models:

- **UniCTokens** (An et al., 2025b): As our foundational architecture, this method employs a multi-stage training protocol. We compare against the original model trained via its standard pipeline to explicitly isolate and verify the performance gains contributed by our proposed Sync-GRPO stage.

- **Yo'chameleon** (Nguyen et al., 2025): A recent unified model capable of personalized understanding and generation. We retrain this model following the specifications in its original study, utilizing a 7B-parameter base model and 1,000 training images per concept.

- **Yo'LLaVA** (Nguyen et al., 2024): A pioneering unified framework addressing personalization via VLMs. Following the original protocol, we train Yo'LLaVA with varying backbone scales (Phi-1.5 (Abdin et al., 2024), 1.3B and Vicuna (Chiang et al., 2023), 13B) to ensure a comprehensive comparison.

- **MC-LLaVA** (An et al., 2025a): Designed to enhance multi-concept personalization, this model utilizes paired textual and visual prompts. It stands as a robust baseline for personalized understanding tasks involving complex concept interactions.

- **RAP-MLLM** (Hao et al., 2025): A retrieval-augmented approach that builds upon RAP-LLaVA. We construct a dedicated personalized database for each concept following the RAP-MLLM framework and post-train on a 260K-sample dataset.

- **Textual Inversion** (Gal et al., 2022): A specialized generative technique that projects visual concepts into the text embedding space, enabling the generation of personalized images via standard text-to-image pipelines.

- **DreamBooth** (Ruiz et al., 2023): A leading fine-tuning method that adapts generative models to user-specific concepts using a small set of reference images. To rigorously assess performance, we evaluate DreamBooth in both low-data (10 images) and high-data (3,000 images) settings.

- **GPT-4o**: We evaluate GPT-4o on our benchmark to establish an empirical upper bound for performance capability.

**Evaluation Metrics.** We adopt a rigorous evaluation protocol covering both personalized understanding and generation tasks. For tasks including personalized recognition, VQA, and standard QA, as well as the fundamental generation Pure Gen. metric in Unifybench, we strictly follow the calculation protocols defined in UniCTokens to ensure direct comparability. For advanced understanding metrics in UnifyBench++), we adopt distinct evaluation protocols for reasoning tasks. **Rea.** is quantified via the BLEU score, calculated between the model's generated rationale and the ground-truth extra_info. For **Dense Rea.**, we employ GPT-4o as an automated judge to assess the logical coherence and factual accuracy of the generated dense reasoning chains. For generation metrics in Unifybench++, we assess generation quality focusing on visual fidelity and text-image alignment. Visual fidelity is measured across all tasks using CLIP-I similarity between the generated output and the user-provided reference concept image. For prompt alignment, we differentiate by task density: (1) Non-dense tasks (**Rea. Gen.**) utilize CLIP-T scores. Crucially, we substitute abstract reasoning placeholders in the prompt with specific descriptive details before computation to ensure ground-truth alignment. (2) Dense tasks (**Dense Gen.** and **Dense Rea. Gen.**) employ a fine-grained dense alignment score, where the prompt is segmented by punctuation. GPT-4o validates whether each segment is accurately represented in the image, yielding an average validity ratio in $[0, 1]$.

*Table 5.* **Performance on Personalized Understanding and Generation Benchmarks.** TP = Text Prompt. IP = Image Prompt.

| Type | Method | Yo'LLaVA | | | MC-LLaVA | | |
|---|---|---|---|---|---|---|---|
| | | Rec | VQA | QA | Rec | VQA | QA |
| | | Weight | Acc | Acc | Weight | BLEU | Acc |
| Und. Only | LLaVA+TP | 0.807 | 0.921 | 0.815 | 0.608 | 0.412 | 0.609 |
| | Yo'LLaVA | 0.932 | 0.918 | 0.897 | 0.829 | 0.657 | 0.691 |
| | MC-LLaVA | 0.953 | 0.935 | 0.924 | 0.901 | 0.692 | 0.736 |
| | Qwen2.5-VL+TP | 0.685 | 0.861 | 0.697 | 0.634 | 0.436 | 0.549 |
| | Yo'LLaVA(Phi-1.5) | 0.805 | 0.627 | 0.713 | 0.726 | 0.525 | 0.591 |
| Unified Model | Chameleon+TP | 0.715 | 0.537 | 0.702 | 0.649 | 0.408 | 0.675 |
| | Yo'Chameleon | 0.832 | 0.591 | 0.734 | 0.753 | 0.610 | 0.658 |
| | Show-o+TP | 0.704 | 0.526 | 0.605 | 0.589 | 0.574 | 0.482 |
| | UniCTokens | 0.865 | 0.602 | 0.725 | 0.742 | 0.617 | 0.692 |
| | Sync-R1 | 0.926 | 0.663 | 0.737 | 0.773 | 0.671 | 0.704 |

| Type | Method | DreamBench | | Yo'LLaVA |
|---|---|---|---|---|
| | | CLIP - I | CLIP - T | CLIP - I |
| Gen. Only | Real Images | 0.873 | - | 0.864 |
| | DreamBooth | 0.692 | 0.297 | 0.645 |
| | DreamBooth | 0.815 | 0.293 | 0.788 |
| | Text inversion | 0.675 | 0.284 | 0.632 |
| Unified Model | Chameleon+TP | 0.612 | 0.168 | 0.579 |
| | Chameleon+IP | 0.594 | 0.171 | 0.499 |
| | Show-o+TP | 0.678 | 0.235 | 0.652 |
| | Yo'Chameleon | 0.807 | 0.212 | 0.771 |
| | UniCTokens | 0.788 | 0.299 | 0.806 |
| | Sync-R1 | 0.808 | 0.323 | 0.822 |

**Additional Results and Analysis.** To validate the robustness of our method beyond our primary benchmarks, we conducted assessments on established datasets specifically designed for pure personalized understanding and generation.

**Understanding Performance.** Table 5 (left) presents the results on the Yo'LLaVA (Nguyen et al., 2024) and MC-LLaVA (An et al., 2025a) datasets. Note that since our current training paradigm focuses on single-concept personalization, we restricted the MC-LLaVA evaluation to the single-concept subset to ensure methodological validity. Our approach consistently outperforms existing state-of-the-art unified models. Notably, Sync-R1 surpasses the previous best method, UniCTokens, by an average margin of **5.3%** across all understanding tasks, demonstrating the efficacy of our reasoning-enhanced optimization.

**Generation Performance.** We further conducted a comprehensive evaluation on Dreambench (Ruiz et al., 2023) and the generation split of the Yo'LLaVA dataset. Our method exhibits superior performance compared to UniCTokens across all metrics. Most significantly, under identical training data constraints, our unified model outperforms the specialized generative baseline, DreamBooth. This indicates that Sync-GRPO successfully bridges the gap between understanding and generation, enabling the synthesis of high-fidelity, context-aware images without compromising model versatility.

## F. Extended Qualitative Results

In this section, we provide additional visual evidence to demonstrate the personalized generation capabilities of **Sync-R1**. We specifically showcase results for two distinct concepts, ⟨f_h⟩ and ⟨butin⟩. As highlighted in Figures 7 and 8, segments requiring high-level inferential reasoning are explicitly marked.

The visualizations reveal two core strengths of our approach: (1) Reasoning-to-Image Fidelity: The model successfully decodes implicit conceptual requirements into concrete visual attributes, effectively bridging the gap between abstract reasoning and pixel synthesis. (2) Dense Prompt Adherence: Our method exhibits robust alignment even when presented with dense, multi-attribute textual descriptions, ensuring that every segmented instruction is accurately manifested in the final output. These results further validate that the Sync-GRPO stage significantly enhances the model's ability to handle complex, knowledge-driven personalization tasks.

*Table 6.* **Ablation study on different reward ensembles.**

| Reward Ensembles | Und. | | | | | | | | | Gen. | | | | |
|---|---|---|---|---|---|---|---|---|---|---|---|---|---|---|
| | Rec. | Rea.* | Dense Rea.* | VQA | | QA | | Pure Gen. | | Dense Gen.* | | Rea. Gen.* | | Dense Rea. Gen.* |
| | Weight | BLEU | GPT | BLEU | GPT | BLEU | GPT | CLIP-T | CLIP-I | GPT | CLIP-I | CLIP-T | CLIP-I | GPT | CLIP-I |
| TIER+BER | 0.838 | 0.245 | 0.494 | 0.582 | 0.583 | 0.594 | 0.644 | 0.298 | 0.739 | 0.329 | 0.636 | 0.311 | 0.784 | 0.326 | 0.743 |
| TIER+DER | 0.849 | 0.239 | 0.487 | 0.593 | 0.588 | 0.594 | 0.639 | 0.297 | 0.733 | 0.323 | 0.638 | 0.306 | 0.779 | 0.321 | 0.738 |
| TIER+BER+DER | 0.852 | 0.252 | 0.491 | 0.585 | 0.591 | 0.603 | 0.631 | 0.306 | 0.748 | 0.327 | 0.640 | 0.318 | 0.787 | 0.331 | 0.741 |
| TIER+BER+DER+FER | 0.859 | 0.250 | 0.503 | 0.592 | 0.606 | 0.604 | 0.652 | 0.308 | 0.765 | 0.337 | 0.645 | 0.324 | 0.801 | 0.353 | 0.756 |

## G. Details of the Synergistic Reward Ensemble

The effectiveness of **Sync-GRPO** relies on a multi-faceted reward ensemble that provides dense feedback for both discrete reasoning and high-fidelity generation. This section elaborates on the mathematical formulation of each reward expert and the empirical calibration of their weights.

### G.1. Reward Component Formulations

We employ four distinct reward experts to supervise the multimodal optimization process, covering logical consistency, cross-modal alignment, structural identity, and facial fidelity.

**Text Inference Evaluation Reward (TIER).** To evaluate the logical consistency of the understanding phase, we utilize ERNIE 3.0 (Sun et al., 2021) to compute semantic embeddings. Let $v_{IR}$ be the embedding of the model's intermediate reasoning output, and $\{v_{E_i}\}_{i=1}^{l}$ be the embeddings of $l$ candidate conceptual information segments. We first compute a similarity vector $\mathbf{s} = [s_1, \ldots, s_l]$, where $s_i = \cos(v_{IR}, v_{E_i})$. The final TIER is derived from the Euclidean distance $d$ between $\mathbf{s}$ and the binary ground-truth vector $\mathbf{y} \in \{0,1\}^l$:

$$R_{\text{TIER}} = \frac{1}{2}(2 - \|\mathbf{s} - \mathbf{y}\|_2) \tag{30}$$

Since $\|\mathbf{s} - \mathbf{y}\|_2$ is bounded within $[0,2]$, $R_{\text{TIER}}$ is normalized to $[0,1]$, where 1 represents perfect logical alignment with the ground-truth knowledge.

**BLIP Evaluation Reward (BER).** To ensure cross-modal semantic alignment, we leverage BLIP-2 (Li et al., 2023) as a vision-language judge. Let $P$ be the text prompt and $I_{\text{gen}}$ be the generated image. BER measures the semantic similarity:

$$R_{\text{BER}} = \cos(\text{Embtext}(P), \text{Embimg}(I_{\text{gen}})) \tag{31}$$

This reward penalizes semantic drift, ensuring that the generated visual content strictly adheres to the provided prompt.

**DINOv2 Evaluation Reward (DER).** For structural identity preservation, we utilize DINOv2 (Oquab et al., 2024), which is sensitive to fine-grained textural and structural features. DER is calculated as the cosine similarity between the features of

the generated image $I_{\text{gen}}$ and the reference image $I_{\text{ref}}$:

$$R_{\text{DER}} = \cos(\Phi_{\text{DINO}}(I_{\text{gen}}), \Phi_{\text{DINO}}(I_{\text{ref}})) \tag{32}$$

This ensures that the personalized concept maintains its unique visual identity across different generated contexts.

**Facenet Evaluation Reward (FER).**    To achieve high-fidelity facial synthesis, we implement a dedicated facial enhancement reward. We first employ MTCNN (Zhang et al., 2016) for face alignment and cropping, followed by Facenet (Schroff et al., 2015; Wang & Deng, 2021) to extract idiosyncratic facial embeddings. The reward is defined as:

$$R_{\text{FER}} = \cos(\text{Embface}(I_{\text{gen}}), \text{Embface}(I_{\text{ref}})) \tag{33}$$

This specific supervisor ensures the preservation of subtle facial features essential for human-centric personalization.

### G.2. Weight Calibration and Ablation Study

The final unified reward is a weighted sum $R_{\text{total}} = \sum w_i R_i$, where $\sum w_i = 1$. We conducted a sensitivity analysis to determine the optimal scaling coefficients, as summarized in Table 6.

**Difficulty-Aware Weighting.**    We observe that understanding tasks (requiring complex discrete reasoning) are significantly more challenging to optimize than generative tasks. To balance the gradient contributions, we adopt a **4:6 ratio** between the understanding component ($w_1$) and the aggregate generative components ($w_2 + w_3 + w_4$).

**Domain-Specific Calibration.**    Based on empirical results in Table 6, we observed that the inclusion of $R_{\text{FER}}$ is crucial for human subjects but redundant for inanimate objects. Furthermore, while generative rewards primarily benefit generation metrics, they also exert a subtle positive influence on understanding through the unified reasoning-loop of Sync-GRPO. The final optimized weights $W = (w_1, w_2, w_3, w_4)$ are set as:

$$W = \begin{cases} (0.4, 0.3, 0.3, 0.0), & \text{for Animals/Objects} \\ (0.4, 0.2, 0.2, 0.2), & \text{for Human Subjects} \end{cases} \tag{34}$$

**Ablation Analysis.**    As shown in Table 6, our "Synergistic Ensemble" (TIER+BER+DER+FER) yields the best overall performance. Notably, even when human concepts constitute only half of the dataset, the fine-grained supervision provided by $R_{\text{FER}}$ leads to a substantial increase in overall generation quality without compromising the model's performance on other categories.

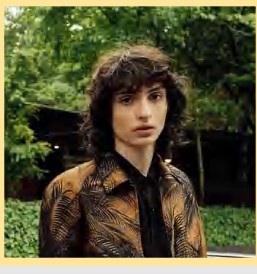

A photo of ‹f_h›.

A photo of ‹f_h› standing in the foreground. Behind him is a road cutting through lush and green rolling hills , with a dreamy atmosphere.

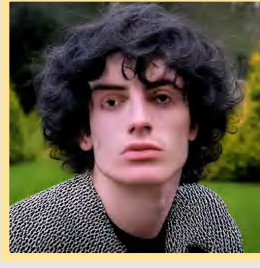

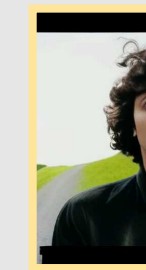

**name:** ‹f_h›
**info:** ‹f_h› is a young man with pale skin, prominent cheekbones, and medium-length curly black hair.
**extra_info:**
1.‹f_h› enjoys hiking in the mountains during autumn.
2.‹f_h› has a pet dog.
3.‹f_h› always went cycling in the countryside in 2019.

A photo of ‹f_h› hiking activity in his favorite season.

A photo of ‹f_h› hiking in the mountains during his favorite season. The background is filled with colorful foliage.

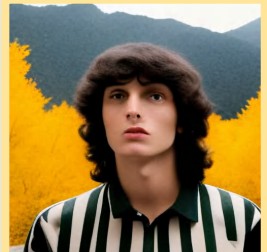

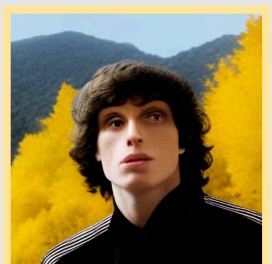

*Figure 7.* **Qualitative Visualization of Concept ⟨f_h⟩.**

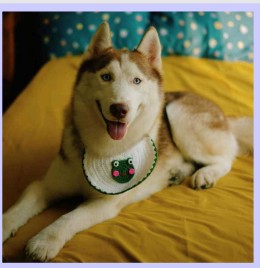

A photo of ‹butin›.

A photo of ‹butin› lying on green grass, with its mouth open and tongue out. In the background, there's a wooden fence and many green trees.

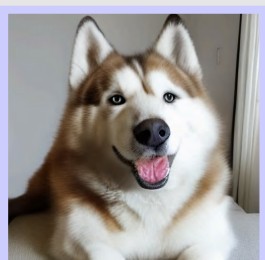

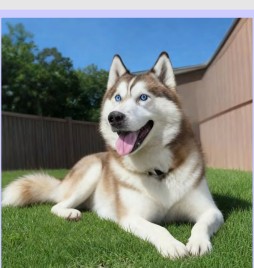

**name:** ‹butin›
**info:** ‹butin› is a Siberian Husky dog with light brown and white fur, blue eyes, and expressive facial features often captured in humorous or relaxed poses.
**extra_info:**
1.‹butin› drinks from a large blue water bowl.
2.‹butin› owns a bright orange rubber toy ball.
3.‹butin› owns a crocheted bib.

A photo of ‹butin› drinking from his bowl.

A photo of ‹butin› wearing his bib, drinking from his bowl, with his toy ball nearby.

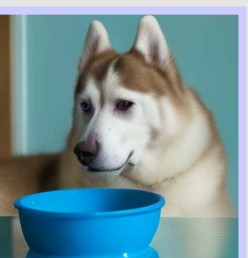

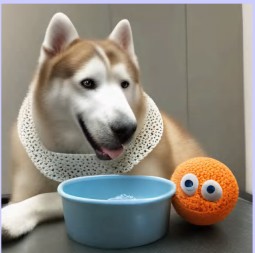

*Figure 8.* **Qualitative Visualization of Concept ⟨butin⟩.**