# OpenReview forum: "Uni-Synergy: Bridging Understanding and Generation for Personalized Reasoning via Co-operative Reinforcement Learning"
_ICML.cc/2026/Conference — Submitted to ICML 2026_

### Official Review · Reviewer_kVhq · 2026-03-04

**Soundness:** 3
**Presentation:** 2
**Significance:** 2
**Originality:** 3
**Overall Recommendation:** 4
**Confidence:** 4

**Summary:**

The paper proposes Sync-R1, a unified reinforcement learning framework that explicitly couples personalized multimodal understanding and generation within a single reasoning trajectory.
Specifically, Sync-GRPO adapts Dr.GRPO to jointly optimize text reasoning tokens and discrete diffusion image tokens under a composite reward ensemble, and Dynamic Group Scaling (DGS) early-filters low-potential trajectories using an intermediate surrogate reward to reduce gradient variance and speed up training.
For evaluation, the authors augment UnifyBench to UnifyBench++ with denser textual descriptions and richer user contexts.
Experiments show that the method outperforms prior baselines, especially in dense, reasoning-heavy settings.

**Compliance With Llm Reviewing Policy:**

Affirmed.

**Final Justification:**

The authors’ response has addressed my concerns, and I think this paper could be accepted.

**Key Questions For Authors:**

1. What does DSOG mean? Is it the same as DGS?
2. Section 4.2, which introduces reward design, may be placed in Section 3.2.

**Limitations:**

The main limitations are mentioned in the Weaknesses section.

**Strengths And Weaknesses:**

**Strengths**
1. The paper tackles an important problem: how to make personalized understanding and generation reinforce each other rather than be optimized as loosely connected objectives. The proposed two-phase reasoning design is intuitive and clearly aligned with this motivation.
2. The experiments are reasonably broad. The paper evaluates standard personalized understanding and generation tasks as well as newly added reasoning-intensive settings in UnifyBench++, and includes ablations on the explicit synergistic loop, DGS, and initialization strategies.
3. The reported gains are promising, especially on the harder UnifyBench++ tasks. The paper also reports that DGS reaches 98% of the baseline peak reward about 1.9× faster, which is a practically relevant efficiency result if robust.

**Weaknesses**
1. While DGS may reduce gradient variance, it also seems to change the effective optimization objective. In GRPO, low-reward samples are still important because they provide useful contrastive signals. Some discarded trajectories may contain valuable failure modes that could help the model correct its most critical weaknesses. It is therefore unclear whether DGS trades improved efficiency for the loss of informative negative supervision.
2. The α_text/α_image weighting schedule is not clearly analyzed by ablation, yet these hyperparameters can significantly bias learning.
3. The UnifyBench++ construction and splits are only partially described. Without a commitment to release, its reproducibility and community value are diminished.

---

> ### Author Rebuttal · Authors · 2026-03-31
>
> We thank the **reviewer kVhq** for recognizing (1) our two-phase reasoning design for coupling understanding and generation, (2) the practical and theoretical value of DGS, and (3) the significance of the UnifyBench++ benchmark.
>
> > W1. Optimization objective shift and trajectory filtering in DGS.
>
> We agree that DGS changes the effective sampling distribution. However, its goal is not to remove all low-reward samples, but to filter trajectories that are already highly predictive of being uninformative at the current optimization stage. In practice, the trajectories discarded by DGS are mostly catastrophic structural failures rather than informative hard negatives.
>
> To test this more directly, we conducted a controlled comparison of selection strategies under a fixed compute budget on a representative concept subset including one human, one animal, and one object (**Table 1** in [the PDF](https://anonymous.4open.science/r/REBUTTAL-D2DF/111111%20(5).pdf)). Besides the default Sync-GRPO baseline and DGS, we compared Random-$k$, Bottom-$k$, and a Top/Bottom Mix, all using the same retained group size ($k=9$) and an expanded candidate pool ($N=36$). Bottom-$k$ performs substantially worse and fails to reach 98\% of the baseline peak reward within the maximum training budget. Top/Bottom Mix also underperforms both Sync-GRPO and DGS. In contrast, DGS achieves the best efficiency-quality trade-off, reaching 1.8$\times$ acceleration while outperforming Sync-GRPO on all reported metrics in this subset.
>
> We also include qualitative examples of discarded and retained trajectories in **Figure 1** of the same PDF. Across representative concepts, discarded samples are typically catastrophic failures with little recoverable semantic signal, whereas retained low-quality samples remain visually plausible but miss fine-grained details. This suggests that DGS mainly removes low-information noise rather than informative failure modes that could provide useful corrective supervision.
>
> > W2. Sensitivity analysis of the text-to-image weighting ratio.
>
> Thank you for pointing this out. We agree that this ablation should have been included more explicitly. We therefore added the sensitivity study in **Table 2** of [the PDF](https://anonymous.4open.science/r/REBUTTAL-D2DF/111111%20(5).pdf), where we vary the weighting between the text-side and image-side policy optimization terms while keeping all other settings fixed.
>
> The results show a clear trade-off. If $\alpha_{\mathrm{image}}$ dominates too strongly (0.2:0.8), reasoning-intensive metrics deteriorate substantially, especially Rea.* and Dense Rea.*. Conversely, if $\alpha_{\mathrm{text}}$ is too large (0.8:0.2), generation-side alignment drops noticeably, especially on Rea. Gen. and Dense Rea. Gen. The adopted setting $\alpha_{\mathrm{text}}:\alpha_{\mathrm{image}} = 0.4:0.6$ gives the best overall balance and achieves the strongest or near-strongest performance across both reasoning and generation metrics. We will include this analysis in the revised version to better justify this hyperparameter choice.
>
> > W3. Reproducibility and release of UnifyBench++.
>
> We agree with this concern. If the paper is accepted, we commit to releasing UnifyBench++, including the split definitions and evaluation scripts, so that the community can reproduce the benchmark and build upon it. We will make this commitment explicit in the revised version.
>
> > Q1. Distinction between DSOG and DGS.
>
> Yes. We apologize for the confusion. "DSOG" is a typo referring to the same module as DGS (Dynamic Group Scaling). "DSOG" was an earlier internal draft name ("Dynamic Scaling of Group Size"), and we inadvertently failed to update the label in that figure. We will correct this typo in the final version. Thank you for catching this.
>
> > Q2. Relocation of Section 4.2.
>
> We fully agree with this suggestion. The synergistic reward ensemble is an integral part of the Sync-GRPO objective, so moving the reward-design discussion closer to the RL objective in the Method section would improve the logical flow and readability. We will adopt this restructuring in the revised version.
>
> We hope these clarifications and the additional evidence help address the reviewer’s concerns and strengthen the overall assessment of our work.

---

> > ### Author Rebuttal · Reviewer_kVhq · 2026-04-01
> >
> > Thanks. I will maintain my positive score. Note that DSOG appears in both Figure 2 and Table 2.

---

> > > ### Author Response · Authors · 2026-04-02
> > >
> > > We appreciate your careful attention to detail again, which helps improve the clarity of our paper. We will rectify this notation inconsistency in the final version of the manuscript. Thank you for maintaining your positive score!

---

### Official Review · Reviewer_rgfS · 2026-03-12

**Soundness:** 3
**Presentation:** 4
**Significance:** 3
**Originality:** 4
**Overall Recommendation:** 4
**Confidence:** 4

**Summary:**

This paper proposes an end-to-end reinforcement learning framework named Sync-R1, which explores the synergy between personalized understanding and generation tasks in Unified Multimodal Models (UMMs). Sync-R1 extracts fine-grained conceptual information from user-provided contexts and subsequently utilizes this information to guide the creation of personalized content. To achieve joint optimization of this process, the authors introduce a group-based reinforcement learning method, Sync-GRPO, along with a composite reward mechanism. Furthermore, to overcome the high computational costs associated with multimodal reinforcement learning, the paper proposes a Dynamic Group Scaling (DGS) strategy. This strategy reduces gradient variance and accelerates convergence by filtering out low-potential trajectories during the early denoising stages. Finally, the authors construct a more challenging benchmark, UnifyBench++, and demonstrate that Sync-R1 achieves state-of-the-art (SOTA) performance across multiple tasks on this benchmark.

**Compliance With Llm Reviewing Policy:**

Affirmed.

**Final Justification:**

The authors have adequately addressed my concerns in the rebuttal. I maintain my original score of weak accept.

**Key Questions For Authors:**

**Q1**: In the joint reward formulation of Equation (8), when the generated image scores highly in visual aesthetics or structural fidelity (e.g., DER/FER) but deviates in text logical reasoning consistency (TIER), how does the Sync-GRPO framework internally balance these conflicting reward signals? During training, did you observe any reward hacking phenomena where the model catered to an easily attainable high-scoring reward?

**Q2**: The DGS strategy performs early truncation and filtering during the initial stages of the diffusion process. How was this specific threshold ratio determined experimentally? If the filtering were performed at earlier or later steps, what would be the specific trade-offs regarding final generation quality and training acceleration ratio?

**Q3**: Could the authors provide some preliminary experiments or discussions exploring the generalization capabilities and potential bottlenecks of this RL paradigm on larger-scale models?

**Limitations:**

yes

**Strengths And Weaknesses:**

**Strengths**

**S1:** Explicitly integrating personalized understanding and generation into a unified reinforcement learning feedback loop is a promising and novel paradigm. This explicit two-stage reasoning mechanism breaks the semantic bottleneck of implicit alignment found in previous multi-task fine-tuning approaches and is highly logically consistent.

**S2:** The proposed Dynamic Group Scaling (DGS) strategy offers excellent practical value and is accompanied by theoretical support for variance reduction. By computing surrogate rewards in the early stages of diffusion generation to filter out low-quality trajectories, it accelerates RL convergence, which is of great significance for computationally expensive multimodal RL training.

**S3:** The introduction of UnifyBench++ provides a more challenging benchmark for the evaluation of large personalized multimodal models. The inclusion of dense text descriptions and rich reasoning contexts better reflects the complex semantic loads encountered in real-world applications.

**Weaknesses**

**W1:** Although the paper proposes a composite reward function, the specific calculation details of each independent reward during the joint RL update, the resolution mechanisms for potential reward conflicts, and the parameter sensitivity analysis are discussed rather briefly in the main text. This leaves readers with a lack of information when evaluating the robustness and anti-reward hacking capabilities of this multimodal reward mechanism.

**W2:** The current experiments primarily rely on the 1.3B parameter Show-o as the base model. While it achieves results superior to some 7B baseline models, it lacks scaling behavior validation for directly applying Sync-R1 on unified multimodal architectures with larger parameter counts. Validating the performance of the DGS strategy and Sync-GRPO in a larger parameter space would strengthen the persuasiveness of the proposed method.

---

> ### Author Rebuttal · Authors · 2026-03-31
>
> We thank the **reviewer rgfS** for the feedback, especially for recognizing: (1) our two-stage reasoning loop for personalization, (2) the value of DGS, and (3) UnifyBench++ as a challenging benchmark.
>
> > W1 / Q1. Reward balancing, robustness, and reward hacking.
>
> Thank you for this important point. We agree that the main text could explain the reward ensemble more clearly. The appendix already provides the detailed formulation of each reward expert, and we will move more of this material into the main paper in the revision. We also add **Table 1** in [the supplementary PDF](https://anonymous.4open.science/r/REBUTTAL-D2DF/111111%20(4).pdf) to summarize the role, input, target behavior, and activation scope of TIER, BER, DER, and FER.
>
> Sync-GRPO does not use a separate arbitration module. Instead, objectives are balanced through a unified trajectory-level reward. In particular, the understanding reward (TIER) has an absolute weight of 0.4, larger than any generation-side reward, so the model cannot simply ignore reasoning and exploit an easier visual reward channel.
>
> We visualize training on `<adrien_brody>` to show the reward dynamics. In **Figure 1**, all reward components increase smoothly, without one rising while others collapse. We also perform leave-one-out training and track the held-out reward. **Figure 2(a)** shows that when TIER is excluded, the held-out TIER still improves, suggesting that the generation-side rewards do not conflict with the understanding objective. **Figure 2(b)** shows that when FER is excluded, the held-out FER remains stable, indicating that FER is complementary rather than competitive.
>
> We also add a sensitivity analysis (**Table 2**). Perturbing the TIER weight by $\pm 15\%$ around the default changes all averaged task scores by less than 1% relatively, showing that the reward mechanism is robust and not fragile to tuning.
>
> > Q2. Selection and trade-offs of the DGS truncation ratio.
>
> Our initial choice of $t \approx 0.2T_{\mathrm{total}}$ was motivated by prior observations that coarse semantic structure in MaskGIT-style iterative generation typically emerges after early denoising. We now include a truncation-step sweep in **Table 3**. The trade-off is clear. If filtering is too early (step 5, ratio 0.10), the surrogate reward correlates poorly with the final reward (Pearson $r=0.34$), hurting generation quality and removing the speed advantage. If filtering is too late (steps 15 or 20), the surrogate becomes slightly more predictive ($r\approx 0.81/0.82$), but much of the computational benefit is lost, reducing acceleration to 1.3× or 1.1×. The adopted setting at step 10 (ratio 0.20) achieves near-best average generation quality while preserving the strongest wall-clock acceleration (1.9×).
>
> In short, earlier truncation gives more speedup but less reliable surrogate ranking, while later truncation gives slightly better fidelity but weaker acceleration. Our chosen setting is the best compromise.
>
> > W2 / Q3. Scaling to larger UMMs and potential bottlenecks.
>
> We agree that larger-scale validation is important. To address this, we additionally train Sync-R1 on Janus-Pro-7B, with results reported in **Table 4**. Compared with its matched SFT baseline, Sync-R1 improves 14 of 15 metrics on Janus-Pro, e.g., Rea. Gen. CLIP-T from 0.287 to 0.331 and Dense Rea. Gen. CLIP-I from 0.683 to 0.779. These results show that the core Sync-GRPO RL paradigm transfers to a larger unified multimodal backbone.
>
> We also clarify the distinction between Sync-GRPO and DGS. Sync-GRPO only requires language and image outputs to be modeled in a discrete decision space, so it naturally extends to tokenized UMMs such as Janus-Pro. DGS, however, is an acceleration module that depends on an intermediate truncation interface during iterative denoising. Therefore, in its current form, DGS is compatible with MaskGIT-/diffusion-style generation, but not with pure autoregressive decoders such as Janus-Pro. The Janus-Pro experiment should thus be interpreted as evidence of scaling for Sync-GRPO without DGS. That said, recent work has begun extending GRPO-style RL beyond discrete-token UMMs to flow/diffusion generators [1,2], suggesting that the broader Sync-GRPO+DGS framework is scalable in principle.
>
> We also agree that scaling this paradigm further introduces practical bottlenecks. The main one is systems cost: GRPO-style online RL requires sampling multiple trajectories per prompt, which becomes increasingly expensive in VRAM, throughput, and distributed communication at 7B scale and beyond.
>
> We hope these clarifications and additional empirical results help address the reviewer’s concerns and strengthen the overall assessment of our work.
>
> [1] DanceGRPO: Unleashing GRPO on Visual Generation
> [2] Flow-GRPO: Training Flow Matching Models via Online RL

---

> > ### Author Rebuttal · Reviewer_rgfS · 2026-04-04
> >
> > Thank you for the comprehensive rebuttal. The additional experiments and clarifications have fully addressed my concerns. I will maintain my positive score.

---

> > > ### Author Response · Authors · 2026-04-04
> > >
> > > We sincerely appreciate your time in reviewing our rebuttal and your continued support of our work. It is very encouraging to know that our new experiments have successfully resolved your questions. We will carefully integrate the extra baseline comparisons and ablation studies into the final manuscript as promised. Thank you once again for your insightful suggestions.

---

### Official Review · Reviewer_HPmr · 2026-03-13

**Soundness:** 2
**Presentation:** 2
**Significance:** 2
**Originality:** 2
**Overall Recommendation:** 4
**Confidence:** 3

**Summary:**

This paper proposes Sync-R1, a reinforcement learning framework designed to jointly optimize personalized understanding and generation in unified multimodal models. The method introduces Sync-GRPO, a reinforcement learning objective that aligns reasoning outputs with generation quality, and Dynamic Group Scaling (DGS) to improve training efficiency by filtering low-quality trajectories. Experiments on the proposed UnifyBench++ benchmark show improved performance on reasoning-intensive multimodal personalization tasks.

**Compliance With Llm Reviewing Policy:**

Affirmed.

**Final Justification:**

The author have increased my main concern, thus increasing the score to weak accept

**Key Questions For Authors:**

How does Sync-R1 perform when Sync-GRPO is applied without any SFT pre-initialization (i.e., directly on a base Show-o model)? This would clarify whether the RL stage is doing the heavy lifting or primarily refining a strong SFT baseline.

**Limitations:**

Yes

**Strengths And Weaknesses:**

**Strengths:**
1. The DGS mechanism has a clear theoretical grounding (Theorem 3.1, Corollary 3.2), and the claimed 1.9× wall-clock speedup is empirically supported.
2. UnifyBench++ meaningfully extends prior benchmarks with reasoning-intensive and dense-text evaluation scenarios that better stress-test personalization.



**Weakness:**
1. All experiments use Show-o (1.3B, 512×512) as the sole backbone. It is unclear whether the synergistic RL approach transfers to other unified architectures (e.g., Chameleon, Janus, Emu3) or scales to larger models. The generality claim is weakened by single-architecture evaluation.
2. The benchmark is constructed by the authors and evaluated largely with GPT-4o as a judge. The reliability and reproducibility of GPT-4o-based scoring for dense reasoning and generation tasks is not validated (e.g., inter-annotator agreement, correlation with human judgment).
3.  The method requires four separate reward models (ERNIE 3.0, BLIP-2, DINOv2, FaceNet) with category-specific weight tuning (different weights for humans vs. objects). This adds substantial implementation complexity and may limit applicability to new domains without careful recalibration.

---

> ### Author Rebuttal · Authors · 2026-03-31
>
> We sincerely thank the **reviewer HPmr** for recognizing the strengths of DGS and the value of UnifyBench++ for reasoning-intensive personalization.
>
> > W1. Single backbone (Show-o) limits generality.
>
> We agree the original submission did not sufficiently support this claim. Sync-GRPO mainly requires a unified multimodal model with discrete tokens across modalities, so reasoning and generation can be optimized within one GRPO trajectory. This already covers a broader family of tokenized unified backbones beyond Show-o. Recent work also suggests that GRPO-style RL can extend to diffusion- and flow-based visual generators [1][2], indicating that the core principle is not tied to one backbone family.
>
> To address this concern directly, we additionally trained Sync-R1 on Janus-Pro-7B. As shown in **Table 1** of [PDF](https://anonymous.4open.science/r/REBUTTAL-D2DF/111111%20(3).pdf), Sync-R1 outperforms the matched UniCTokens-SFT baseline on 14/15 metrics. For example, Rea.-Gen. CLIP-T improves from 0.287 to 0.331, and Dense Rea.-Gen. CLIP-I from 0.683 to 0.779. This shows the gains are not specific to Show-o or the 1.3B scale. We will include this experiment in the revision.
>
> > W2. Reliability of GPT-4o evaluation.
>
> We agree judge reliability must be validated carefully. At the same time, LLM/VLM-based evaluation is increasingly used when target properties are too compositional or fine-grained for conventional metrics; for example, ConceptMix [3] uses GPT-4o in its evaluation pipeline.
>
> More importantly, we directly validated our pipeline via a cross-judge and human study in **Table 2** of [PDF](https://anonymous.4open.science/r/REBUTTAL-D2DF/111111%20(3).pdf). For GPT-4o-evaluated task groups, we additionally report Gemini-2.5-pro and human scores. For each task group with more than 200 samples, we randomly sampled 200 instances. Trends are highly consistent: GPT-4o preserves exactly the same model ranking as human evaluation across all five overlapping task groups (Spearman rank correlation = 1.0 in every case), while Gemini shows the same ranking on four groups and only a minor swap on the remaining one. Sync-R1 remains the strongest method under GPT-4o, Gemini, and human judgment. We will add this analysis to the revision.
>
> > W3. Complexity and applicability of the reward design.
>
> The four reward experts correspond to standard personalization desiderata. TIER, BER, and DER are shared across all categories; only FER is category-specific, enabled only for human subjects because facial identity supervision is not applicable to animal/object concepts. Detailed reward roles are summarized in **Table 3** of [PDF](https://anonymous.4open.science/r/REBUTTAL-D2DF/111111%20(3).pdf).
>
> Importantly, our weighting rule is coarse and structured rather than heavily hand-tuned. We fix the overall understanding:generation ratio at 4:6, and within generation the active reward terms are equally weighted. Thus, moving to a new category does not require per-concept tuning.
>
> We also added a sensitivity analysis in **Table 4** of [PDF](https://anonymous.4open.science/r/REBUTTAL-D2DF/111111%20(3).pdf). Perturbing the understanding weight $w_{\mathrm{Und.}}$ by $\pm 15\%$ causes only very small changes: all averaged task scores vary by less than 1% relatively, and the model ranking remains unchanged. This suggests Sync-R1 is not fragile to exact coefficients and should not require delicate recalibration in nearby domains, consistent with recent visual RL work using fixed multi-reward designs or simple reward aggregation [4][5].
>
> > Q. Performance of Sync-GRPO without SFT.
>
> Our new results in **Table 5** of [PDF](https://anonymous.4open.science/r/REBUTTAL-D2DF/111111%20(3).pdf) show that SFT initialization is important, but RL is not merely a minor polishing step.
>
> Applying Sync-GRPO directly to the base Show-o model without SFT initialization causes severe degradation relative to the full Sync-R1 pipeline, indicating that SFT plays an essential bootstrapping role. At the same time, RL does more than refine a strong SFT baseline: Sync-R1 consistently improves both the basic Joint-SFT and the stronger UniCTokens-SFT baselines across all evaluated metrics. Thus, SFT provides the necessary foundation, while Sync-GRPO unlocks higher-level gains in cross-modal alignment and reasoning-intensive personalization.
>
> We hope these clarifications and new results help address the reviewer’s concerns.
>
> [1] DanceGRPO: Unleashing GRPO on Visual Generation
> [2] Flow-GRPO: Training Flow Matching Models via Online RL
> [3] ConceptMix: A Compositional Image Generation Benchmark with Controllable Difficulty
> [4] T2I-R1: Reinforcing Image Generation with Collaborative Semantic-level and Token-level CoT
> [5] DiffusionNFT: Online Diffusion Reinforcement with Forward Process

---

> > ### Author Rebuttal · Reviewer_HPmr · 2026-04-04
> >
> > I thank the author for the rebuttal, I have increased the score to weak accept

---

> > > ### Author Response · Authors · 2026-04-04
> > >
> > > Thank you very much for reviewing our rebuttal and for raising the score. We deeply appreciate your time and the constructive feedback, which have significantly helped us improve the quality and rigor of our paper. We ensure that the new experiments will be carefully integrated into the final revision. Thank you again for your support and guidance throughout the review process!

---

### Official Review · Reviewer_5SMQ · 2026-03-13

**Soundness:** 3
**Presentation:** 1
**Significance:** 2
**Originality:** 2
**Overall Recommendation:** 4
**Confidence:** 4

**Summary:**

This paper proposes Sync-R1, a unified reinforcement learning framework that couples personalized understanding and generation through a co-operative reasoning loop. By orchestrating a two-phase trajectory where the model extracts conceptual knowledge to guide content creation, the framework enables comprehension and generation torefine each other within an integrated reward landscape.

**Compliance With Llm Reviewing Policy:**

Affirmed.

**Final Justification:**

The author address some of my concerns. I will raise the score to weak accpet.

**Key Questions For Authors:**

Please refer to Weaknesses 1-6 for detailed concerns.

**Limitations:**

The paper contains incomplete figures with missing visual content or placeholders (e.g., Figures 1, 2, and 6), which hinders a clear understanding of the proposed framework

**Strengths And Weaknesses:**

Strengths:

1. The Dynamic Group Scaling (DGS) strategy effectively reduces computational overhead by filtering low-potential trajectories in early denoising stages.
2. The inclusion of training dynamics and reward curves provides clear transparency into the learning process. These visualizations demonstrate how the adaptive threshold and pass rate evolve.


Weaknesses:

1. The paper may contain incomplete figures with missing visual content (e.g., Figures 1, 2, and 6). This omission hinders a clear understanding of the proposed framework and prevents a visual assessment of the generation quality, reflecting inadequate paper preparation.
2. Sync-GRPO primarily integrates the existing GRPO algorithm with four off-the-shelf models (ERNIE 3.0, BLIP-2, DINOv2, and Facenet) serving as reward experts. This approach resembles system-level engineering rather than a fundamental algorithmic contribution.
3. The reward weighting strategy relies on rigid, hard-coded coefficients, such as setting weights to (0.4, 0.2, 0.2, 0.2) for humans and (0.4, 0.3, 0.3, 0.0) for animals or objects. These manually defined, static rules lack generalizability and are susceptible.
4. The experimental evaluation lacks comparisons with standard contemporary baselines in personalized generation, such as BAGEL. Relying primarily on comparisons with UniCTokens and Show-o is insufficient to substantiate the claims of state-of-the-art performance.
5. The Dynamic Group Scaling mechanism proposes training acceleration via early-stage filtering around the 10th denoising step. However, it fails to account for the significant computational and communication latency introduced by repeatedly invoking external VLMs (e.g., BLIP-2) for intermediate scoring, which may offset the theoretical efficiency gains.
6. The evaluation is strictly limited to personalized tasks (UnifyBench++) and neglects to address catastrophic forgetting. Because reinforcement learning fine-tuning can degrade foundational capabilities, it is essential to evaluate the model on standard, non-personalized benchmarks (e.g., COCO, MMMU) to confirm that general VQA and generation performance remains intact.

---

> ### Author Rebuttal · Authors · 2026-03-31
>
> We sincerely thank **reviewer 5SMQ** for the assessment and for recognizing the efficiency of DGS and the transparency of our training visualizations.
>
> Anonymous supplementary material: [Anon PDF](https://anonymous.4open.science/r/REBUTTAL-D2DF/111111%20(2).pdf)
>
> > W1. Incomplete figures (Figs. 1, 2, 6).
>
> We sincerely apologize for this issue. After re-checking the submission, we verified that Figures 1, 2, and 6 render correctly in the OpenReview web viewer on our side, suggesting a PDF rendering/compatibility issue rather than missing content. We include the rendered versions in the Anon PDF and will ensure the revised version no longer suffers from this issue. In addition, Figure 2 contains a typo ("DSOG" instead of "DGS"), which has also been corrected.
>
> > W2. Novelty of Sync-GRPO.
>
> We respectfully clarify that Sync-GRPO is not merely a system-level integration of reward experts. Its core novelty is an R1-like unified reasoning framework: (i) a two-phase explicit reasoning loop that first performs personalized understanding and then uses inferred concept knowledge to guide generation, and (ii) a unified optimization objective whose policy ratio spans both reasoning tokens and image-generation tokens, so understanding and generation are optimized jointly within a single trajectory rather than as disjoint objectives. The reward experts only provide complementary supervision (e.g., alignment and identity); see **Table 2** in the Anon PDF.
>
> > W3. Rigid reward weights and generalizability.
>
> We appreciate the reviewer’s scrutiny of our reward design. The coefficients are not arbitrary. As detailed in **Appendix G.2**, the 4:6 split between understanding and generation is grounded in a difficulty-aware prior reflecting their relative optimization difficulty. The only category-specific change is FER: its weight is set to zero for animals/objects because facial fidelity is semantically inapplicable there, not because of fragile per-concept tuning. Detailed reward roles are given in **Table 2** of the Anon PDF.
>
> To address susceptibility, we perturbed the overall understanding weight by $\pm 15\%$ around the default setting. As shown in **Table 1** of the Anon PDF, all metrics remain highly stable, with relative changes strictly below 1\%. This shows that Sync-R1 is robust to moderate coefficient variation and does not rely on fragile numerical tuning.
>
> Heuristic multi-reward combinations also remain common in recent RL studies. For instance, [1] averages reward scores, [2] uses equal reward weights, and [3] optimizes FLUX.1-dev with a fixed 1:0.26 ratio. Our sensitivity analysis provides direct evidence that this design is likewise robust in our framework.
>
> > W4. Insufficient baseline comparisons (e.g., BAGEL).
>
> The original submission already included broader comparisons in **Appendix E**, including personalized-understanding baselines (e.g., Yo'LLaVA), personalized-generation baselines (e.g., DreamBooth), and unified baselines (e.g., Chameleon). We additionally evaluated Janus-Pro-7B and BAGEL-7B on UnifyBench++. As shown in **Table 3** of the Anon PDF, Sync-R1, despite using a much smaller 1.3B model, achieves the best performance on 10 of the 15 reported metrics, showing that the gains of Sync-GRPO remain competitive against recent 7B unified models.
>
> > W5. Overhead of invoking external models.
>
> We agree that real wall-clock cost is the correct measure. The 1.9× speedup reported in **Figure 5** is already measured against wall-clock time rather than training steps, so it includes the latency of the external scoring pipeline, although we did not state this clearly enough.
>
> To clarify this, we added a runtime breakdown in **Table 4** of the Anon PDF. For rejected candidates, the intermediate surrogate evaluation after the first 10 denoising steps adds only about 0.32s, while allowing DGS to bypass about 9.03s of subsequent decoding and final reward evaluation. Thus, the surrogate check adds little overhead relative to the computation it saves.
>
> > W6. Lacks general catastrophic forgetting evaluation.
>
> To verify that the personalized RL stage does not degrade general-purpose capabilities, we evaluated catastrophic forgetting on GenEval, MMMU, POPE, VQAv2 (COCO), and COCO-FID (randomly selected 1k samples).
>
> As shown in **Table 5** of the Anon PDF, Sync-R1 remains essentially unchanged relative to both the base model and UniCTokens-SFT. These results indicate that the personalized RL stage does not cause meaningful catastrophic forgetting, while still delivering clear gains on personalized reasoning and generation.
>
> We hope these clarifications and the newly added empirical results help resolve the reviewer’s concerns and improve the overall assessment of our work.
>
> [1] T2I-R1: Reinforcing Image Generation with Collaborative Semantic-level and Token-level CoT
> [2] DiffusionNFT: Online Diffusion Reinforcement with Forward Process
> [3] TempFlow-GRPO: When Timing Matters for GRPO in Flow Models

---

> > ### Author Rebuttal · Reviewer_5SMQ · 2026-04-06
> >
> > My concerns have been adequately addressed.

---

> > > ### Author Response · Authors · 2026-04-07
> > >
> > > We sincerely thank the reviewer for the highly encouraging feedback and for confirming that our rebuttal has fully addressed your concerns. Your insightful comments have greatly helped us improve the quality and clarity of our manuscript. We are fully committed to incorporating the additional experiments (e.g., the BAGEL baseline and foundational capability evaluations) and the completed figures (Figures 1, 2, and 6) into the final camera-ready version. Thank you again for your time, constructive suggestions, and support for our work!

---

### Decision · Program_Chairs · 2026-04-30

**Decision:**

Reject

**Comment:**

After reviewing the paper, the four reviewer reports, and the authors’ rebuttal, I recommend Reject. The reviewers initially raised substantial concerns regarding missing figure content, limited architectural generality, reliance on heuristic reward weighting, insufficient baseline comparisons, and lack of evaluation for catastrophic forgetting. In their rebuttal, the authors provided additional experiments (e.g., results on Janus-Pro-7B, BAGEL comparisons, sensitivity analyses, and forgetting evaluations) and clarified that the figure issues stemmed from PDF rendering. While the rebuttal addressed several technical points and three reviewers raised their scores to weak accept, one reviewer (5SMQ) retained significant concerns about the core novelty being system-level engineering rather than fundamental algorithmic contribution, the static reward weights lacking generalizability, and the potential overhead of external models offsetting DGS gains. The authors’ responses, though thorough, did not fully resolve these foundational weaknesses. Given the remaining concerns about novelty, robustness, and practical applicability, the paper does not meet the bar for acceptance at ICML 2026.